# ASSESS: A SEMANTIC AND STRUCTURAL EVALUATION FRAMEWORK FOR STATEMENT SIMILARITY

**Xiaoyang Liu**[*] **Tao Zhu**[*] **Zineng Dong** **Yuntian Liu** **Qingfeng Guo**
**Zhaoxuan Liu** **Yu Chen** **Tao Luo**[†]
School of Mathematical Sciences, Shanghai Jiao Tong University
{xiaoyang.liu, branden2004, stju_dzn, fulcrums, gracegqf,
liuzhaoxuan, lcly2462525, luotao41}@sjtu.edu.cn

## ABSTRACT

Despite significant strides in statement autoformalization, a critical gap remains in the development of automated evaluation metrics capable of assessing formal translation quality. Existing metrics often fail to balance semantic and structural information: string-based methods neglect semantics, whereas proof-based approaches offer no graded similarity when proofs fail. To address these issues, we introduce **ASSESS** (*A Semantic and Structural Evaluation Framework for Statement Similarity*), which captures syntactic structure by transforming formal statements into operator trees and computes a real-valued similarity score using our novel **TransTED** (*Transformation Tree Edit Distance*) **Similarity** metric by incorporating semantic transformations. For rigorous validation, we present **EPLA** (*Evaluating Provability and Likeness for Autoformalization*), a benchmark comprising 1,247 expert-annotated formal statement pairs derived from miniF2F and ProofNet, distinctively labeled for both semantic provability and structural likeness. Experiments on the EPLA benchmark demonstrate that TransTED Similarity surpasses existing methods, achieving state-of-the-art accuracy and Kappa score. The benchmark dataset, code, and detailed experimental results are available at https://github.com/XiaoyangLiu-sjtu/ASSESS.

## 1 INTRODUCTION

Formal languages such as Isabelle (Paulson, 1994), HOL Light (Harrison, 1996), Rocq (Barras et al., 1997), and Lean (de Moura et al., 2015; Moura & Ullrich, 2021) have recently gained prominence within the mathematical community for their capacity to rigorously verify proofs. Nevertheless, formalizing mathematical content is a labor-intensive process that demands substantial time, effort, and a profound familiarity with these specialized languages and their corresponding mathematical libraries, such as Mathlib (mathlib Community, 2020). Consequently, the task of autoformalization (Szegedy, 2020), defined as translating theorem statements and proofs from natural language into their formal counterparts, has become an active area of research.

While autoformalization has advanced rapidly, methods for evaluating its output have lagged behind. Existing metrics, which generally function by assigning a similarity score, are constrained by a fundamental trade-off between capturing semantic meaning and preserving structural information. String-based metrics such as BLEU (Papineni et al., 2002) rely on surface-level n-gram overlap, rendering them sensitive to inconsequential lexical variations while remaining blind to underlying semantic content. Conversely, proof-based approaches (Li et al., 2024; Liu et al., 2025a) can guarantee semantic equivalence but are limited by prover brittleness, resulting in high false negatives and no graded feedback for unproven statements. Additionally, syntax-based metrics (Jiang et al., 2023) operate merely at the level of grammatical compliance, offering negligible semantic or structural insights. Finally, the LLM-as-a-Judge (Ying et al., 2024a) approach, while powerful, introduces prohibitive costs and significant reproducibility concerns. These collective shortcomings highlight

---

[*]Equal contribution.
[†]Corresponding author. Also affiliated to Institute of Natural Sciences, MOE-LSC, CMA-Shanghai, Shanghai Jiao Tong University.

a critical gap: the need for an automated evaluation metric that robustly integrates semantic and structural information to assess statement similarity in a reproducible and efficient manner.

In this work, we propose **ASSESS** (*A Semantic and Structural Evaluation Framework for Statement Similarity*), a novel two-stage framework for evaluating formal statement pairs. The core of ASSESS is its novel metric, **TransTED** (*Transformation Tree Edit Distance*) **Similarity**. It is designed to capture both semantic and structural nuances, offering a reproducible solution that relies exclusively on CPU resources. In the first stage, ASSESS leverages the Lean Language Server to parse each formal statement pair into their operator trees (OPTs), a representation that captures hierarchical structure more effectively than raw text. To overcome the semantic limitations of a standard Tree Edit Distance (TED) (Zhang & Shasha, 1989), the second stage augments the TED with a curated set of transformations. This augmentation enables TransTED Similarity to robustly measure statement similarity where purely structural or semantic methods fail.

To enable rigorous evaluation, we introduce **EPLA** (*Evaluating Provability and Likeness for Autoformalization*). We constructed this benchmark by autoformalizing informal statements from the miniF2F-test (Zheng et al., 2022) and ProofNet-test (Azerbayev et al., 2023) datasets, utilizing a combination of two domain-specific and two general-purpose models. We filtered the candidate formalizations using the Lean compiler to ensure syntactic validity. The surviving instances were paired with ground-truth references and expert-annotated based on semantic provability and structural likeness. The benchmark contains 1,247 annotated pairs in total, with 831 from EPLA-miniF2F and 416 from EPLA-ProofNet.

Experimental results demonstrate that TransTED Similarity consistently outperforms all baselines. It establishes a new state-of-the-art by achieving 70.16% accuracy and a 0.35 Kappa score on EPLA-miniF2F, alongside 67.31% accuracy and a 0.30 Kappa score on EPLA-ProofNet. Compared to existing approaches, our metric provides a more robust assessment than brittle proof-based methods, is more reproducible than LLM-based judges, and surpasses the performance of n-gram-based metrics. Furthermore, a detailed ablation study identifies the transformation component as the key factor in these performance gains. This confirms that our mechanism bridges the gap between semantic equivalence and structural likeness by dynamically aligning syntactically diverse yet logically identical expressions.

We summarize our main contributions as follows:

1. We propose a novel method that leverages the Lean Language Server to automatically parse formal statements into operator trees. This structured representation enables the direct application of TED Similarity to quantify statement similarity.
2. We introduce TransTED Similarity, which augments the traditional TED Similarity by incorporating semantic transformations into the distance computation. This semantic-aware metric achieves state-of-the-art performance on the EPLA benchmark.
3. We present EPLA, a comprehensive benchmark tailored for this evaluation task, comprising 1,247 formal statement pairs annotated by domain experts.

## 2 RELATED WORK

**Autoformalization.** The goal of autoformalization is to translate informal mathematics into formal code. Research in this area, particularly for theorem statements, has evolved significantly. While early work often relied on neural machine translation (Wang et al., 2018; Cunningham et al., 2022), the transformative impact of Large Language Models (LLMs) has given rise to three dominant strategies. These include exploring the efficacy of few-shot prompting (Wu et al., 2022; Agrawal et al., 2022; Zhou et al., 2024); improving capabilities by fine-tuning LLMs (Gao et al., 2025; Lu et al., 2024; Liu et al., 2025b; Wu et al., 2025; Yu et al., 2025) on relevant parallel statements; and integrating retrieval-augmented generation (Zhang et al., 2024) to achieve enhanced performance.

**Automated Evaluation.** Automated evaluation in the context of autoformalization denotes metrics that assign quantitative scores to machine-generated formal statements, with the goal of estimating their similarity relative to a natural-language source (Weng et al., 2025). Early efforts relied on grammatical validity (Jiang et al., 2023) and string similarity (Azerbayev et al., 2023), yet these often struggle with semantic understanding. While FormalAlign (Lu et al., 2025) integrates evaluation

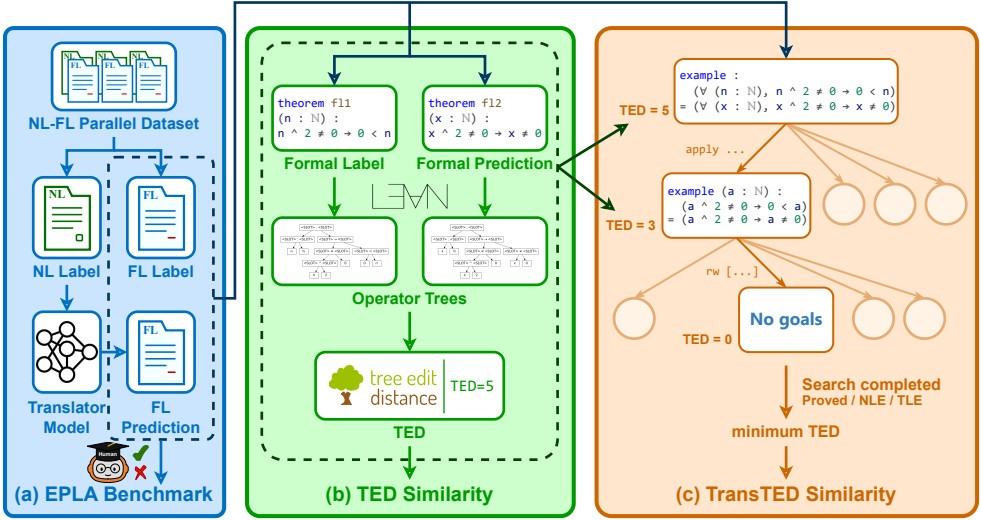

Figure 1: Overview of the ASSESS Framework. (a) EPLA Benchmark: This dataset is constructed using four distinct translators to generate Formal Language (FL) pairs, followed by compilation checks and human evaluation. (b) TED Similarity: A baseline metric computed by converting FL pairs into operator trees to calculate the Tree Edit Distance (TED) similarity. (c) TransTED Similarity: A novel metric that reformulates an FL pair as an equality. It conducts a tree search guided by specific tactic commands, utilizing TED as a heuristic to minimize the edit distance. The search halts upon satisfying any of the following conditions: successful proof generation (Proved), Node Limit Exceeded (NLE), or Time Limit Exceeded (TLE).

with autoformalization, its internal scoring mechanism cannot serve as a standalone metric. Simultaneously, cross-provability (Murphy et al., 2024; Li et al., 2024; Liu et al., 2025a; Poiroux et al., 2025) between formal statements emerges as a widely accepted standard for automated evaluation, but its effectiveness is constrained by the current progress in automated theorem proving. GTED (Liu et al., 2025c) pioneered the use of operator trees for this evaluation task; however, it suffered from implementation instabilities and its transformation mechanism was confined to simple variable renaming. In contrast, our framework formalizes these concepts into a rigorous pseudometric space and introduces proof-based transformations to capture deep semantic alignment.

**Operator Trees.** Operator trees (OPTs) represent mathematical expressions as syntax trees, with operators as internal nodes and operands as leaves (Zanibbi et al., 2002). Compared with sequence-based formula representations, OPTs explicitly capture the hierarchical structure and semantic relations within expressions, preserving operator precedence and operand dependencies (Zanibbi & Blostein, 2012; Hu et al., 2013). These properties have made OPTs foundational to applications in mathematical information retrieval (MIR). For instance, systems extract structural features from OPTs for similarity searching (Zhong & Zanibbi, 2019) or combine them with other representations for formula retrieval, as in the MCAT system (Kristianto et al., 2016). The utility of OPTs also extends to deep learning, where models like MathBERT (Peng et al., 2021) integrate OPT structures during pretraining to enhance semantic understanding , and encoders such as FORTE (Wang et al., 2021) learn formula representations directly from OPT-based inputs.

## 3 METHODOLOGY

This section details our proposed framework, ASSESS, as illustrated in Figure 1. We begin by establishing the theoretical foundations in Section 3.1. Subsequently, we present the two core metrics: the basic TED Similarity in Section 3.2, followed by our primary contribution, the semantically-aware TransTED Similarity in Section 3.3.

### 3.1 PRELIMINARIES

Our framework quantifies statement similarity by computing the distance between their structural representations. We model these representations as vertices within a weighted, undirected graph, where dissimilarity is captured by the shortest-path distance between them. To formalize this graph-based approach, we begin with the definition of a pseudometric space.

A **pseudometric space** is a set $X$ equipped with a function $d : X \times X \to \mathbb{R}_{\geq 0}$, called a pseudometric, that satisfies the following axioms for all $x, y, z \in X$:

- Identity: $d(x, x) = 0$
- Symmetry: $d(x, y) = d(y, x)$
- Triangle inequality: $d(x, z) \leq d(x, y) + d(y, z)$

Crucially, a pseudometric differs from a metric in that $d(x, y) = 0$ does not necessarily imply $x = y$. This property is essential for our application, as structurally distinct formal statements can be semantically equivalent (e.g., $a + b$ and $b + a$).

**Definition 1** (Shortest-path distance). *Let $G = (V, E, w)$ be an undirected, weighted graph, where $V$ is the set of vertices, $E$ is the set of edges, and $w : E \to \mathbb{R}_{\geq 0}$ is the edge weight function. The shortest path distance $d : V \times V \to \mathbb{R}_{\geq 0}$ is defined as:*

$$d(u, v) := \inf \left\{ \sum_{e \in p} w(e) \,\middle|\, p \text{ is a path from } u \text{ to } v \text{ with finitely many edges} \right\}.$$

By convention, $d(v, v) = 0$. This function $d$ satisfies the required axioms, confirming that $(V, d)$ constitutes a pseudometric space.

### 3.2 TED SIMILARITY

Our baseline metric, TED Similarity, quantifies structural correspondence between two formal statements. This is achieved in three steps: (1) representing each statement as a hierarchical operator tree (OPT); (2) computing the Tree Edit Distance (TED) between the two OPTs; and (3) normalizing this distance to produce a final similarity score.

#### 3.2.1 OPT CONSTRUCTION

To capture the hierarchical structure of formal statements, we represent each as a labeled, ordered operator tree. We leverage the Lean Language Server to parse a formal statement, from which we derive the tree's topology based on the nested scopes of its elements: operators become parent nodes and their arguments become ordered children. During construction, we apply two additional transformations to standardize the tree structure:

- **Placeholder Representation:** We append a placeholder `<SLOT>` to the labels of all non-leaf (operator) nodes. This explicitly marks the node's functional role, disambiguating operators from operands that might share the same name.
- **Parentheses Omission:** Parentheses enforce operator precedence in a linear string representation. Since a tree's topology inherently encodes this hierarchy, parentheses are redundant structural artifacts and are omitted from the OPT.

As an example, Figure 2 demonstrates the constructed OPT for the following formal statement:
```
theorem eq : ((Σ x ∈ Finset.range 10, (x + 1) ^ 2) % 10 = 5) = ((Σ
 n ∈ Finset.Icc 1 9, n ^ 2) % 10 = 5) := by sorry
```

#### 3.2.2 METRIC CALCULATION

With formal statements converted to OPTs, we can instantiate the pseudometric space defined in Section 3.1. The set of vertices $X$ becomes the set of all possible OPTs, and the distance function $d$ is the Tree Edit Distance.

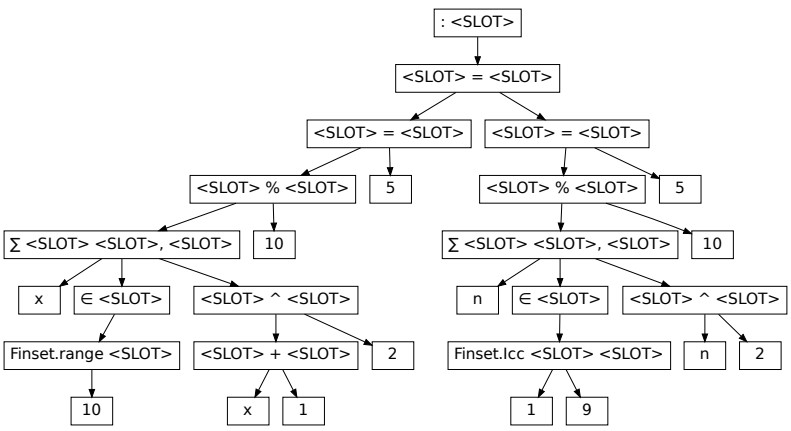

Figure 2: The operator tree (OPT) of a formal statement.

**Definition 2** (Tree Edit Distance). *Let $X$ be the set of all labeled, ordered operator trees (OPTs). The Tree Edit Distance (TED), $d_{TED} : X \times X \to \mathbb{R}_{\geq 0}$, between two trees $T_1, T_2 \in X$ is the minimum total cost of a sequence of operations that transforms $T_1$ into $T_2$. The standard operations and their associated non-negative costs are:*

- **Deleting** *a node, connecting its children to its parent in order. Cost: $c_{del}$.*

- **Inserting** *a node, the inverse of a delete operation. Cost: $c_{ins}$.*

- **Relabelling** *a node by changing its label. Cost: $c_{rel}$.*

*For $d_{TED}$ to be a valid pseudometric, the costs must satisfy $c_{del} = c_{ins}$, ensuring symmetry.*

To enable meaningful comparisons across trees of varying sizes, we define TED Similarity by normalizing the absolute Tree Edit Distance relative to tree size.

**Definition 3** (TED Similarity). *The TED Similarity between two OPTs, $T_1$ and $T_2$, is:*

$$\text{sim}_{TED}(T_1, T_2) := 1 - \frac{d_{TED}(T_1, T_2)}{\max\{|T_1|, |T_2|\}},$$

*where $|T|$ is the number of nodes in tree $T$.*

This formulation assumes unit costs for all operations ($c_{\text{del}} = c_{\text{ins}} = c_{\text{rel}} = 1$), a standard convention we adopt (Zhang & Shasha, 1989). The denominator $\max\{|T_1|, |T_2|\}$ serves as a normalization factor, representing the cost of deleting all nodes in the larger tree.

### 3.3    TRANSTED SIMILARITY

While TED Similarity effectively captures structural differences, it is syntactically rigid and penalizes semantically equivalent expressions, such as $a + b$ and $b + a$. We therefore introduce TransTED Similarity, which integrates semantic awareness into the distance computation via transformations.

#### 3.3.1    THEORETICAL FRAMEWORK

In our framework, we define a concept, transformation, to integrate semantic awareness into distance functions. We start with an example: $i = j$ implies that $f(i) = f(j)$. In other words, $f(i) = f(j)$ is no stronger than $i = j$. We say that the contrast of $f(i)$ and $f(j)$ can be transformed to the contrast of $i$ and $j$. Generally, a weaker contrast can be transformed to a stronger one. We notice that the two sides of a weak contrast are usually near in semantic. Therefore the distance between the original contrast should be no larger than that between the transformed one. For compatibility with TED, we will define a new distance between OPTs instead of expressions.

Here we provide a formal definition of TransTED. Let $X$ be the set of all operator trees. We define a new pseudometric $d^*$ on $X$ that is constrained by the following two properties:

- Domination by TED: For any two trees $T_1, T_2 \in X$, the new distance is bounded by the original Tree Edit Distance: $d^*(T_1, T_2) \leq d_{\text{TED}}(T_1, T_2)$.
- Monotonicity under transformation: If a pair of formal expressions $(e_x, e_y)$ can be transformed into a logically stronger pair $(e_u, e_v)$ (i.e., if $e_u = e_v$ implies $e_x = e_y$), then the metric must satisfy $d^*(\text{OPT}(e_x), \text{OPT}(e_y)) \leq d^*(\text{OPT}(e_u), \text{OPT}(e_v))$.

Finding the largest pseudometric $d^*$ that satisfies these conditions can be formulated as a linear programming problem. The following theorem establishes that this problem has a unique optimal solution, which we define as **TransTED**. The full proof is provided in Appendix C.

**Theorem 1.** *Let $X$ be an arbitrary set, $b : X \times X \to \mathbb{R}_{\geq 0}$ be a function and $T$ is a subset of $(X \times X)^2$. Consider the function space*

$$
\mathcal{F} := \left\{ f \,\middle|\, f \text{ is a pseudometric on } X, \text{ and } \begin{cases} \forall x, y \in X, f(x, y) \leq b(x, y), \\ \forall((x, y), (u, v)) \in T, f(x, y) \leq f(u, v) \end{cases} \right\}.
$$

*Then there exists a unique maximum function $\overline{f} \in \mathcal{F}$ such that for any $(x, y) \in X \times X$,*

$$
\overline{f}(x, y) = \sup_{f \in \mathcal{F}} f(x, y).
$$

### 3.3.2 IMPLEMENTATION

Since practical implementations are constrained by a finite set of transformations, our algorithm computes a tractable upper bound of TransTED rather than its exact theoretical value. Crucially, however, if the transformation sequence establishes semantic equivalence, the computed distance becomes exactly 0, coinciding with the true theoretical value.

Table 1: Additional Tactic Commands

| Tactic command | Original Goal | Transformed Goal |
|---|---|---|
| `apply congrArg` | `f x = f y` | `x = y` |
| `apply congrFun` | `f x = g x` | `f = g` |
| `apply forall_congr;`
`intro _` | `(a → b) = (a → c)` | `(_ : a) ⊢ b = c` |
| `apply implies_congr;`
`all_goals (try rfl)` [1] | `(a → c) = (b → d)` | `a = b` or `c = d` or None |
| `ext` | `(fun (x : A) => f x) =`
`(fun (y : A) => g y)` | `(x : A) ⊢ f x = g x` |
| `rw [propext and_imp]` | `a ∧ b → c` | `a → b → c` |
| `norm_cast` [2] | - | - |

In the context of the Lean theorem prover, transformation can be conceptualized as the set of tactics that adhere to our theoretical framework (see Theorem 1). For implementation, on one hand, we curated a set of tactic commands, including `rw?`, the most related automatic tactic for searching equivalence transformations, and additional ones selected for their practical efficacy, listed in Table 1. On the other hand, since these tactics operate on single equalities rather than pairs of statements, we first construct an equation as the initial contrast to be transformed and unified later, by connecting a given pair of formal statements with an equal sign. Concrete examples are provided in Appendix F.3.

---

[1] At least one of `a = b` and `c = d` should be proved by `rfl`, or otherwise the tactic command generates two new goals, which is forbidden in our heuristic search.

[2] `norm_cast` undertakes simple type coercions.

To circumvent the combinatorial explosion of applying all tactics, we apply a search algorithm. The algorithm performs a heuristic search through transformations. The guiding heuristic prioritizes tactics that reduce the TED between OPTs on the left- and right-hand sides of the equation, effectively pruning less promising branches. Searching is terminated as long as one of the following conditions is satisfied: The equivalence is proved, the node limit is exceeded (NLE), and the time limit is exceeded (TLE). This approach allows our method to efficiently find short-distance transformation paths and the complete algorithm is detailed in Appendix D. Finally, we normalize the raw TransTED distance into a TransTED Similarity score, defined in an analogous way as Definition 3.

## 4 EXPERIMENTS

### 4.1 THE EPLA BENCHMARK

We introduce **EPLA** (*Evaluating Provability and Likeness for Autoformalization*), a benchmark designed to transcend the limitations of current evaluation protocols. Existing benchmarks frequently rely on coarse binary labels (e.g., provable vs. unprovable), precluding a nuanced assessment of metric performance. For instance, a slightly incorrect yet easily fixable translation is significantly more valuable than a completely erroneous one, yet binary metrics classify both identically as False. EPLA provides a more nuanced evaluation standard to address this critical gap.

EPLA is built upon two established datasets: miniF2F-test (Zheng et al., 2022) and ProofNet-test (Azerbayev et al., 2023). We utilize the specific versions of these datasets provided by Numina[3] for miniF2F-test and DeepSeek[4] for ProofNet-test. To generate candidate formalizations, we translate natural language statements from the datasets into Lean 4 using four distinct models: two domain-specific models, Herald Translator (Gao et al., 2025) and Goedel-Formalizer-V2-8B (Lin et al., 2025), and two general-purpose models, Gemini-2.5-Pro (Comanici et al., 2025) and Qwen3-Max (Yang et al., 2025). The prompts used for the domain-specific models are consistent with the original papers, while the prompts used for the general-purpose models are provided in Appendix F.5.

Following the application of a compilation filter to retain only syntactically valid statements, a panel of seven experts with mathematical backgrounds annotated and cross-validated each pair of formal statements. The annotation methodology employs three Boolean sub-labels: **Provability**, representing the semantic equivalence of the statements; **Likeness before transformation**, indicating structural similarity between the original statements; and **Likeness after transformation**, indicating structural similarity following semantic-preserving transformations. The resulting EPLA benchmark, detailed in Table 2, comprises 1,247 annotated instances. Further specifications regarding the benchmark's format and label definitions are provided in Appendix E.

Table 2: Statistics of the EPLA benchmark

|  | Herald Translator | Goedel-Formalizer | Gemini 2.5 Pro | Qwen3-Max | Total |
|---|---|---|---|---|---|
| EPLA-miniF2F | 198 | 234 | 175 | 224 | 831 |
| EPLA-ProofNet | 89 | 130 | 62 | 135 | 416 |

### 4.2 EXPERIMENT SETTING

**Baselines.** We benchmark TED Similarity and TransTED Similarity against several competing baseline methods to validate the efficacy of our proposed metric. To ensure fair and consistent evaluation across these methods, we establish specific conventions for handling theorem names. For string-based approaches (e.g., Identity Match, BLEU), we standardize theorem names in both ground truth and predicted formal statements to `thm`. For proof-based approaches (e.g., Definitional Equality, BEq), we designate the ground truth theorem names as `thm_P` and the predicted theorem names as `thm_Q`. For all other methods, theorem names remain unaltered.

- Identity Match: A predicted formal statement is considered correct if, after removing all whitespace, it is identical to the ground truth.

---

[3]https://huggingface.co/datasets/AI-MO/minif2f_test
[4]https://github.com/deepseek-ai/DeepSeek-Prover-V1.5/tree/main/datasets

Table 3: **Overall results of the competing baselines and our metrics**. The boldface refers to the highest score and the underline indicates the next best result of the metrics. Detailed results about the number of TP, TN, FP and FN are available in Appendix F.2.

| Metric | EPLA-miniF2F | | | | EPLA-ProofNet | | | |
| --- | --- | --- | --- | --- | --- | --- | --- | --- |
| | Precision | Recall | Accuracy | Kappa | Precision | Recall | Accuracy | Kappa |
| *(Baselines)* | | | | | | | | |
| Identity Match | **100.00%** | 8.79% | 32.61% | 0.05 | **100.00%** | 3.69% | 43.51% | 0.03 |
| BLEU | 82.25% | **73.94%** | 68.96% | 0.26 | 72.30% | 43.85% | 57.21% | 0.18 |
| Majority Voting | 91.00% | 31.27% | 46.93% | 0.14 | 66.67% | 45.08% | 54.57% | 0.12 |
| Definitional Equality | 97.09% | 32.57% | 49.46% | 0.19 | 86.21% | 10.25% | 46.39% | 0.07 |
| BEq | 98.60% | 45.77% | 59.45% | 0.29 | 98.77% | 32.79% | 60.34% | 0.28 |
| *(Ours)* | | | | | | | | |
| TED Similarity | 84.51% | 71.99% | 69.56% | 0.31 | 66.11% | 81.56% | 64.67% | 0.23 |
| **TransTED Similarity** | 87.97% | 69.06% | **70.16%** | **0.35** | 68.49% | **81.97%** | **67.31%** | **0.30** |

- BLEU: We follow the implementation in ProofNet (Azerbayev et al., 2023).
- Majority Voting: Following Lean Workbook (Ying et al., 2024a), we employ DeepSeek-V3.2-Exp (DeepSeek-AI, 2025) with temperature $0.7$ for 16 rounds of majority voting.
- Definitional Equality (Liu et al., 2025a): A predicted formal statement is considered correct if `example: thm_P = thm_Q := by rfl` succeeds.
- BEq (Liu et al., 2025a): This metric assesses the mutual provability between a ground-truth formal statement `thm_P` and a predicted formal statement `thm_Q`. It employs InternLM2-Math-Plus-20B (Ying et al., 2024b) with an expanded set of tactics to attempt proofs in both directions. The prediction is considered correct only if both proofs are successful.

**Implementation.** The prompts employed in the Majority Voting baseline are detailed in Appendix F.5. Regarding tactic usage for the BEq baseline, we adopt the normal set of tactics `exact`, `exact?`, `have`, `apply`, `cases'`, `constructor`, `ext`, `intro`, `intros`, `rw`, `use` and fix the number of LLM generation attempts at $k = 16$, consistent with the optimal hyperparameters reported in the original work. For TransTED Similarity, we utilize the top-5 suggestions generated by the `rw?` tactic at each step of the search process and define the termination criteria as a maximum tree size of 32 or a search timeout of 10 minutes.

To ensure a fair comparison with baselines requiring binary labels, we establish a unified evaluation protocol. First, we utilize the provability labels from EPLA as the ground-truth labels for the formal statement pairs. Second, for real-valued metrics such as BLEU, TED Similarity, and TransTED Similarity, we convert scores into binary predictions via thresholding. We report the optimal Kappa score and associated statistics, determined by sweeping over all possible decision thresholds.

All experiments were conducted on the EPLA benchmark using the Lean toolchain `v4.9.0-rc1`. The computational environment consisted of one NVIDIA A800 GPU with 80GB of memory and two Intel Xeon Silver 4216 CPUs, providing a total of 32 physical cores. With the exception of the GPU-dependent BEq baseline, all evaluations were executed on CPUs. Our proposed metric is particularly lightweight, requiring only interaction with the Lean Language Server and REPL.

### 4.3 EXPERIMENT RESULTS

**Overall Comparison.** Table 3 presents a comprehensive overview of our proposed TransTED Similarity compared to various baseline metrics across the EPLA-miniF2F and EPLA-ProofNet benchmarks. Our primary evaluation focuses on accuracy and the Cohen's Kappa, as accuracy directly reflects the overall correctness of the evaluation, and Kappa (Cohen, 1960) offers a robust measure of agreement beyond chance. The results demonstrate that TransTED Similarity achieves state-of-the-art performance across both datasets. On EPLA-miniF2F, it secures the highest accuracy (70.16%) and Kappa (0.35). It maintains this superior performance on the EPLA-ProofNet

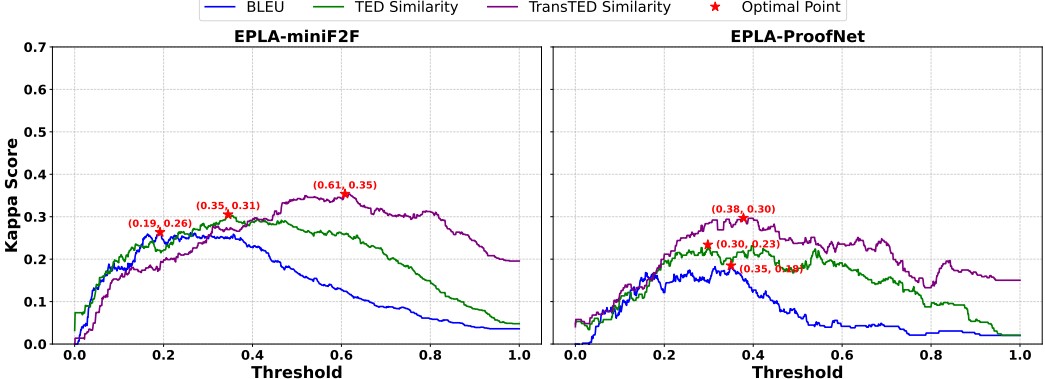

Figure 3: Comparison of BLEU, TED Similarity, and TransTED Similarity across varying decision thresholds on EPLA-miniF2F and EPLA-ProofNet. Red stars indicate the optimal threshold configuration for each metric. Details regarding threshold selection are provided in Appendix F.1.

dataset, again leading with the highest accuracy (67.31%) and Kappa (0.30). This consistent lead, particularly in the Kappa metric, highlights our method's robust ability to align with ground-truth judgments, outperforming all evaluated baselines.

**Comparison with Baselines.** As detailed in Table 3, TransTED Similarity effectively mitigates key limitations inherent to existing baseline categories. First, while proof-based metrics like Definitional Equality and BEq achieve high precision, their performance is plagued by low recall. This brittleness stems from the limitations of current Automated Theorem Provers, which frequent fail to verify valid equivalences, resulting in a high rate of false negatives. Our method sidesteps this rigid dependency, yielding a significantly more balanced assessment (see Appendix B for a comprehensive discussion). Furthermore, TransTED Similarity significantly outperforms the Majority Voting baseline, achieving superior accuracy (70.16% vs. 46.93% on EPLA-miniF2F) and a substantially higher Kappa score (0.35 vs. 0.14). Crucially, our method circumvents the primary limitations of LLM-based metrics: it is GPU-independent and fully reproducible. Finally, our method overcomes a critical limitation of n-gram-based metrics like BLEU: their inability to capture the semantics of mathematical text. As illustrated in Figure 3, TransTED Similarity demonstrates significant robustness, maintaining high stability across a broad spectrum of decision thresholds. Further details regarding threshold selection are provided in Appendix F.1.

**Ablation Study.** We conduct an ablation study comparing our TransTED Similarity metric against the basic TED Similarity metric. The results, presented in Table 3, demonstrate that this component provides a significant performance boost on both datasets. On EPLA-miniF2F, its inclusion increases accuracy by 0.60 percentage points, improving it from 69.56% to 70.16%, and raises the Kappa score from 0.31 to 0.35. This trend holds on EPLA-ProofNet, where accuracy improves by 2.64 percentage points from 64.67% to 67.31%, and the Kappa score increases from 0.23 to 0.30. These consistent gains substantiate our central hypothesis: by integrating semantic transformations, our metric effectively discerns semantic equivalence amidst syntactic diversity. This capability enables TransTED Similarity to deliver a more accurate and human-aligned evaluation, succeeding where purely structural methods, such as the standard TED Similarity, fall short.

**Analysis of Tactic Commands.** To assess the practical utility of our tactic set, we conducted a detailed statistical analysis of their usage patterns during the search process. Specifically, we quantified the frequency with which each tactic command was adopted. Here, an *adopted* tactic is defined as one applied along the optimal transformation path that yields a minimum TED state. These statistics are summarized in Figure 4, with raw data in Appendix F.2.

High usage frequency of `rw?` and `norm_cast` indicates their utility as general-purpose tools for simplification through rewriting and type-coercion normalization, and their low adoption rate is

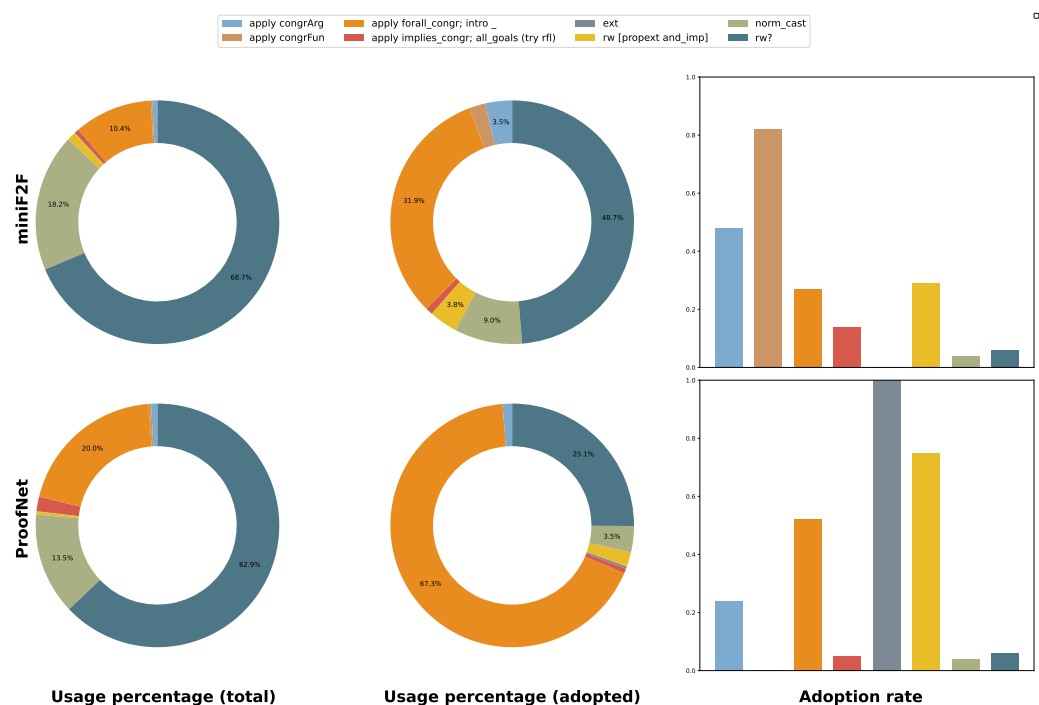

Figure 4: Statistic of the usage frequencies and adoption rates of tactic commands. (A) Usage Percentage (total): The proportion of each tactic command's successful applications out of all successful applications during the entire search process. (B) Usage Percentage (adopted): The proportion of each tactic command's adoption counts out of all adoption counts. (C) Adoption Rate: The proportion of each tactic command's adoption counts out of its total application counts.

consistent with their role in exploring a wide search space and handling routine transformations. Conversely, high adoption rates of `apply` forall_congr; `intro` _ and `rw` [propext and_imp] illstrate their high specificity: they are applied less often, but when their preconditions - decomposing a universal quantifier or restructuring a nested implication - are met, they are highly effective and frequently reside on the critical path towards simplification. Other tactics commands are rarely successfully used, due to their strict application requirements and strong dependence on precise type unification, limiting their general applicability.

Overall, the synergy between high-frequency, general-purpose tactics and their specialized, high-specificity counterparts forms a robust strategy. The former ensures extensive exploration of the search space, while the latter precisely resolves the intricate logical dependencies central to establishing semantic equivalence.

## 5 CONCLUSION

In this work, we addressed the critical gap in evaluating autoformalized statements, where existing metrics fail to adequately balance semantic and structural information. We introduced ASSESS, a novel framework whose core contribution is the TransTED Similarity metric. By augmenting the structural comparison of operator trees with a set of semantic transformations, our method provides a more holistic and robust measure of statement similarity. We also developed EPLA, an expert-annotated benchmark for this task. Experiments conducted on EPLA demonstrate that TransTED Similarity establishes a new state-of-the-art, outperforming existing methods in both accuracy and Kappa score. This work provides the autoformalization community with a more reliable, efficient, and reproducible metric.

## ACKNOWLEDGEMENTS

This work is sponsored by the National Key R&D Program of China Grant No. 2022YFA1008200 (T. L.). We also thank Shanghai Institute for Mathematics and Interdisciplinary Sciences (SIMIS) for their financial support. This research was funded by SIMIS under grant number SIMIS-ID-2025-ST. The authors are grateful for the resources and facilities provided by SIMIS, which were essential for the completion of this work. We appreciate the valuable assistance of Yifan Bai and Yantao Li with the human evaluation.

## REPRODUCIBILITY STATEMENT

We are committed to ensuring the reproducibility of our research. To this end, we provide comprehensive details of our benchmark, experimental setup, and evaluation methodology. The code, benchmark and experimental results are available at `https://github.com/XiaoyangLiu-sjtu/ASSESS`.

Specific details for reproducing our results can be found in the following sections of the paper:

- EPLA Benchmark: The construction methodology, data sources, generation process, and the complete expert annotation scheme for our benchmark are detailed in Section 4.1.
- Baselines and Experiment Setting: The Lean toolchain version, implementation details and specific configurations for all baseline methods are provided in Section 4.2 and Appendix F.5.
- Detailed Results: In addition to the main results in Table 3, detailed per-metric performance, including the number of True Positives, True Negatives, False Positives, and False Negatives, are available in Appendix F.2.

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

## A    THE USE OF LARGE LANGUAGE MODELS (LLMS)

We utilized a large language model (LLM) as a general-purpose writing assistant during the preparation of this manuscript. The primary role of the LLM was to polish the text written by the authors, such as improving grammar. All intellectual contributions, including the research methodology, experimental results, and their interpretation, are entirely the work of the authors. The LLM's function was strictly that of an assistive tool for improving the quality of the written presentation.

## B    LIMITATIONS AND FUTURE WORK

We analyzed some failure cases for understanding TransTED Similarity metric's boundaries and guiding future research. We categorize the primary failure modes into two types.

**Limited Tactic Commands (False Negatives).**    TransTED Similarity relies on a curated set of tactic commands (specifically `rw?` and 7 structural tactics). In complex cases requiring creative leaps or intermediate lemma synthesis, the heuristic search may fail to find a path, leading to a low score for equivalent statements.

---

**EPLA-miniF2F #25**

**NL:** Let $S = 2010 + 2011 + \cdots + 4018$. Compute the residue of $S$, modulo 2009. Show that it is 0.

**Label:**
```
theorem mathd_numbertheory_353 (s : ℕ) (h₀ : s = Σ k in
Finset.Icc 2010 4018, k) : s % 2009 = 0 := by sorry
```

**Prediction:**
```
theorem my_favorite_theorem : (Finset.sum (Finset.Icc 2010
4018) id) % 2009 = 0  := by sorry
```

**TED Similarity:** 0.17
**TransTED Similarity:** 0.17

---

In this case, the current tactic set could not bridge the semantic gap between the variable definition s in the hypothesis and its direct usage in the conclusion. This highlights the need to further optimize the configuration of tactic commands.

**Semantic-Structural Tension (False Positives).**    Because TransTED Similarity includes a structural similarity component, it can assign high scores to statements that are structurally similar but semantically distinct (e.g., off-by-one errors or index shifts).

---

**EPLA-ProofNet #360**

**NL:** Prove that $\left(\sum_{j=1}^{n} a_j b_j\right)^2 \leq \left(\sum_{j=1}^{n} j a_j^2\right)\left(\sum_{j=1}^{n} \frac{b_j^2}{j}\right)$ for all real numbers $a_1, \ldots, a_n$ and $b_1, \ldots, b_n$.

**Label:**
```
theorem exercise_6_3 {n : ℕ} (a b : Fin n → ℝ) : (Σ i, a i ⋆ b
i) ^ 2 ≤ (Σ i : Fin n, i ⋆ a i ^ 2) ⋆ (Σ i, b i ^ 2 / i) := by
sorry
```

**Prediction:**
```
theorem test_problem (n : ℕ) (a b : Fin n → ℝ) : (Σ j : Fin n,
 a j ⋆ b j)^2 ≤ (Σ j : Fin n, (j.val + 1) ⋆ (a j)^2) ⋆ (Σ j :
Fin n, (b j)^2 / (j.val + 1)) := by sorry
```

**TED Similarity:** 0.51
**TransTED Similarity:** 0.81

---

**The final transformed statement:**

```
theorem node_4 (a : ℕ) (a_1 a_2 : Fin a → ℝ) :(Σ i : Fin a, ↑↑i
 * a_1 i ^ 2) * Σ i : Fin a, a_2 i ^ 2 / ↑↑i = (Σ j : Fin a, (↑
↑j + 1) * a_1 j ^ 2) * Σ j : Fin a, a_2 j ^ 2 / (↑↑j + 1) := by
 sorry
```

The label (left hand side) iterates over $i$, while the prediction (right hand side) iterates over $j + 1$. Although TransTED Similarity fails to provide a correct binary verification on mathematical equivalence due to structural likeness, the searching result precisely locates the semantic difference. This suggests that TransTED Similarity might be highly effective for human-in-the-loop debugging, filtering nearly-correct candidates for expert review.

Furthermore, we concede that TransTED Similarity does not guarantee 100% precision. This is a fundamental characteristic of continuous metrics, which require a decision threshold and thus incur a precision-recall trade-off. Unlike rigorous decision metrics like BEq (which guarantee 100% precision but suffer from extremely low recall due to prover brittleness), TransTED Similarity is designed as a fractionalized metric. Its goal is to maximize alignment with human judgment by balancing precision and recall, rather than strictly filtering for absolute logical provability. Consequently, TransTED Similarity's false positives are rarely meaningless. Instead, they are typically samples with "high semantic similarity but minor technical discrepancies," such as:

- Consistency issues in variable naming or type.
- Differences in implicit type parameters.
- The omission of a single minor hypothesis.

In practical scenarios like human-in-the-loop evaluation or dataset pre-screening, these similar but not correct candidates are highly valuable. A metric that flags these as highly similar helps human experts locate and correct minor errors, whereas a strict prover like BEq would simply reject them, providing no feedback.

## C  PROOF OF THEOREM 1

Here we provide the full proof of the Theorem 1.

*Proof.* $0 \in \mathcal{F}$ so $\mathcal{F} \neq \emptyset$. Let $\widehat{f} : X \times X \to \mathbb{R}_{\geq 0}$ be a function defined as

$$\widehat{f}(x, y) = \sup_{f \in \mathcal{F}} f(x, y).$$

Since $f(x, y) \leq b(x, y)$ for any $f \in \mathcal{F}$, $\widehat{f}(x, y) \leq b(x, y) < \infty$. It suffices to show that $\widehat{f} \in \mathcal{F}$. By definition of $\widehat{f}$, for any point pairs $(x, y) \in X \times X$ and any neighbourhood $\mathcal{N}$ of $\widehat{f}(x, y)$, there exists $f_{x,y,\mathcal{N}} \in \mathcal{F}$ such that

$$f_{x,y,\mathcal{N}}(x, y) \in \mathcal{N}.$$

Now we check that $\widehat{f}$ satisfies these properties one by one.

First, we verify that $\widehat{f}$ is a pseudometric. Take any $x, y, z \in X$. Since for any $f \in \mathcal{F}$, $f(x, x) = 0$,

$$\widehat{f}(x, x) = \sup_{f \in \mathcal{F}} f(x, x) = \sup_{f \in \mathcal{F}} 0 = 0.$$

Similarly, for any $f \in \mathcal{F}$, $f(x, y) = f(y, x)$, so

$$\widehat{f}(x, y) = \sup_{f \in \mathcal{F}} f(x, y) = \sup_{f \in \mathcal{F}} f(y, x) = \widehat{f}(y, x).$$

To show that the triangle inequality holds, we introduce an arbitrary neighbourhood $\mathcal{N}$ of $\widehat{f}(x, z)$. Then

$$\mathcal{N} \ni f_{x,z,\mathcal{N}}(x, z) \leq f_{x,z,\mathcal{N}}(x, y) + f_{x,z,\mathcal{N}}(y, z) \leq \widehat{f}(x, y) + \widehat{f}(y, z).$$

Let $\mathcal{N}$ is an arbitrary neighbourhood of $\widehat{f}(x,z)$, we have $\widehat{f}(x,z) \leq \widehat{f}(x,y) + \widehat{f}(y,z)$.

Next, we show that $\widehat{f}$ satisfies the extra requirements in the definition of $\mathcal{F}$. We have shown that $\widehat{f}(x,y) \leq b(x,y)$. Now take any $((x,y),(u,v)) \in T$. Notice that for any $f \in \mathcal{F}, f(x,y) \leq f(u,v)$. Now for any neighbourhood $\mathcal{N}$ of $\widehat{f}(x,y)$, we have

$$\mathcal{N} \ni f_{x,y,\mathcal{N}}(x,y) \leq f_{x,y,\mathcal{N}}(u,v) \leq \widehat{f}(u,v),$$

hence $\widehat{f}(x,y) \leq \widehat{f}(u,v)$. $\qquad\qquad\square$

## D   PSEUDOCODE FOR TRANSTED

This section provides the detailed pseudocode for our TransTED algorithm. The implementation consists of two primary functions: a primary wrapper TRANSTED that handles initial checks, and the core TEDAFTERTRANSFORMATION function, which executes the heuristic search. The search (`line 13`) is guided by using the Tree Edit Distance as its heuristic (`line 18`). If a proof of equivalence is found, the distance is 0; otherwise, if the time limit is exceeded, the algorithm returns the smallest TED found among all visited nodes (`line 21`).

---

**Algorithm 1** TransTED

---

1:  **function** TRANSTED(FL1, FL2)
2:      eq $\leftarrow$ (FL1 = FL2)
3:      **if** Compile(eq) fails **then**
4:          **return** TED(eq)
5:      **end if**
6:      **return** TEDAFTERTRANSFORMATION(eq)
7: **end function**

8:  **function** TEDAFTERTRANSFORMATION(eq)
9:      **if** Completed(UseTactic(eq, `exact?`)) **then**
10:         **return** 0
11:     **end if**
12:     HeuristicSearch(
13:         start : eq,
14:         stop : the goal completed $\vee$ node limit exceeded (NLE) $\vee$ time limit exceeded (TLE)
15:         search method : suggestions by `rw?` and some given additional tactic commands
16:         valid nodes: a single goal of equality, or "completed"
17:         heuristic function : TED between the expressions trees of both sides of the equality
18:     )
19:     **if** TLE **then**
20:         **return** smallest TED of all the visited nodes
21:     **else**
22:         **return** 0                                 ▷ The goal is completed within the time limit
23:     **end if**
24: **end function**

---

## E   BENCHMARK STRUCTURE

The EPLA data format is structured as a tuple: (natural language statement, ground-truth formalization, candidate formalization, annotation label). The annotation label is further decomposed into three Boolean sub-labels, each designed to quantify a specific dimension of similarity and address a distinct application scenario.

| Sub-label | Definition | Application Scenario |
|---|---|---|
| Provability | Whether the two statements are semantically equivalent. | To evaluate if a metric correctly assesses the semantic equivalence of two statements. |
| Likeness before transformation | Whether the two (original) statements are structurally similar. | To evaluate if a metric reflects the editing cost required to make one statement structurally identical to another. |
| Likeness after transformation | Whether the two statements are structurally similar after semantic-preserving transformations | To evaluate if a metric reflects the editing cost required to make one statement semantically identical to another. |

Due to inherent logical dependencies between sub-labels, the label space is constrained to the five valid combinations. The underlying logic is governed by two key implications:

- **Provability $\implies$ Likeness after transformation:** Since provable statements are inherently semantically equivalent, a valid transformation path exists by definition, satisfying the likeness criterion.

- **Likeness before transformation $\implies$ Likeness after transformation:** Structural similarity serves as a strict condition; the editing cost required to demonstrate semantic equivalence is naturally upper-bounded by the cost required to achieve structural identity.

| Provability | Likeness before transformation | Likeness after transformation |
|---|---|---|
| True | True | True |
| True | False | True |
| False | True | True |
| False | False | True |
| False | False | False |

# F  EXPERIMENTAL DETAILS AND CASE STUDIES

This appendix supplements the main paper with comprehensive experimental details and qualitative case studies. We begin by outlining the methodology for threshold selection in Appendix F.1, followed by the detailed experimental results in Appendix F.2. We then provide a qualitative assessment of metric performance, presenting illustrative examples of TransTED Similarity in Appendix F.3 and discussing false positives in Definitional Equality and BEq in Appendix F.4. Finally, the specific prompt templates used for autoformalization and evaluation are documented in Appendix F.5.

## F.1  THRESHOLD SELECTION

We observe that the choice of threshold is inherently benchmark-dependent. This dependency stems from the richness of the underlying mathematical formalism: plentiful mathematical structures $\Rightarrow$ more structurally variant expressions of the same meaning $\Rightarrow$ lower possible structural likeness.

For instance, the optimal TransTED Similarity threshold for EPLA-miniF2F (0.61) is significantly higher than that for EPLA-ProofNet (0.38). This disparity reflects the greater structural complexity and variation inherent to the latter. Consequently, we recommend that practitioners calibrate thresholds according to the structural diversity of their specific domain, using the optimal values reported in this work as reference baselines. To ensure robustness, we further advise evaluating model performance across a spectrum of thresholds rather than relying on a single static cut-off.

## F.2 DETAILED EXPERIMENTAL RESULTS

Tables 6 and 7 present the experimental results on the EPLA-miniF2F and EPLA-ProofNet benchmarks, respectively. For each metric, we report a detailed classification breakdown (TP, TN, FP, FN) alongside computational costs, measured by total and average execution times. Additionally, Figures 5 and 6 analyze the tactic command counts for TransTED Similarity.

Table 6: Detailed experimental results of automated evaluation metrics on EPLA-miniF2F.

| Metric | EPLA-miniF2F | | | | | | | | | |
|---|---|---|---|---|---|---|---|---|---|---|
| | TP | TN | FP | FN | Precision | Recall | Accuracy | Kappa | Tot. Time (s) | Avg. Time (s) |
| *(Baselines)* | | | | | | | | | | |
| Identity Match | 54 | 217 | 0 | 560 | **100.00%** | 8.79% | 32.61% | 0.05 | 0.1240 | 0.0001 |
| BLEU | 454 | 119 | 98 | 160 | 82.25% | **73.94%** | 68.95% | 0.26 | 10.6390 | 0.0128 |
| Majority Voting | 192 | 198 | 19 | 422 | 91.00% | 31.27% | 46.93% | 0.14 | 11343.1519 | 13.6500 |
| Definitional Equality | 200 | 211 | 6 | 414 | 97.09% | 32.57% | 49.46% | 0.19 | 215.8885 | 0.2598 |
| BEq | 281 | 213 | 4 | 333 | 98.60% | 45.77% | 59.45% | 0.29 | 26133.5967 | 31.4484 |
| *(Ours)* | | | | | | | | | | |
| TED Similarity | 442 | 136 | 81 | 172 | 84.51% | 71.99% | 69.55% | 0.31 | 998.8283 | 1.2020 |
| **TransTED Similarity** | 424 | 159 | 58 | 190 | 87.97% | 69.06% | **70.16%** | **0.35** | 42028.5440 | 50.5759 |

Table 7: Detailed experimental results of automated evaluation metrics on EPLA-ProofNet.

| Metric | EPLA-ProofNet | | | | | | | | | |
|---|---|---|---|---|---|---|---|---|---|---|
| | TP | TN | FP | FN | Precision | Recall | Accuracy | Kappa | Tot. Time (s) | Avg. Time (s) |
| *(Baselines)* | | | | | | | | | | |
| Identity Match | 9 | 172 | 0 | 235 | **100.00%** | 3.69% | 43.51% | 0.03 | 0.1179 | 0.0003 |
| BLEU | 107 | 131 | 41 | 137 | 72.30% | 43.85% | 57.21% | 0.18 | 5.7977 | 0.0139 |
| Majority Voting | 110 | 117 | 55 | 134 | 66.67% | 45.08% | 54.57% | 0.12 | 5371.1336 | 12.9114 |
| Definitional Equality | 25 | 168 | 4 | 219 | 86.21% | 10.25% | 46.39% | 0.07 | 114.6867 | 0.2757 |
| BEq | 80 | 171 | 1 | 164 | 98.77% | 32.79% | 60.34% | 0.28 | 14872.7313 | 35.7518 |
| *(Ours)* | | | | | | | | | | |
| TED Similarity | 199 | 70 | 102 | 45 | 66.11% | 81.56% | 64.67% | 0.23 | 549.1361 | 1.3200 |
| **TransTED Similarity** | 200 | 80 | 92 | 44 | 68.49% | **81.97%** | **67.31%** | **0.30** | 54343.4060 | 130.6332 |

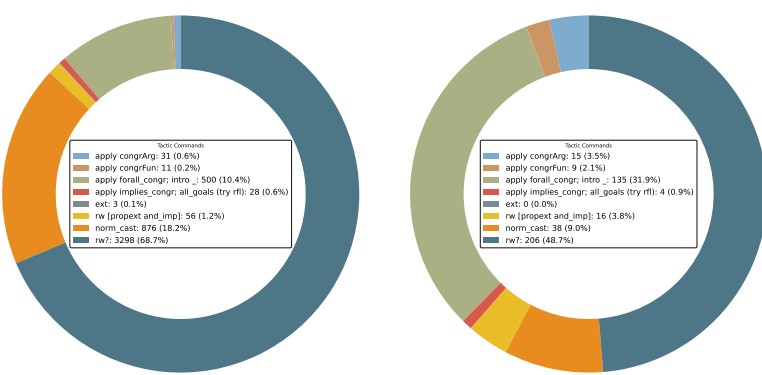

Figure 5: Statistic of the tactic commands in EPLA-miniF2F

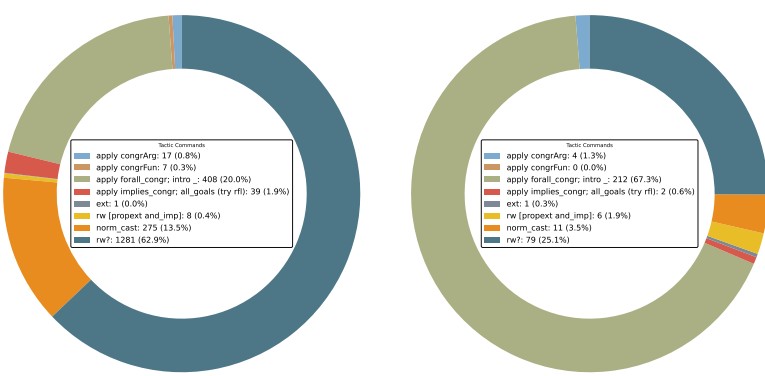

Figure 6: Statistic of the tactic commands in EPLA-ProofNet

### F.3 ILLUSTRATIVE EXAMPLES OF TRANSTED SIMILARITY

This section provides a set of illustrative examples to offer a qualitative understanding of TransTED Similarity's performance, particularly in cases where TED Similarity fails. Each example is presented in a consistent format, including the original natural language statement (NL), the ground-truth formalization (Label), and the model's output (Prediction). We then present the scores from both metrics, the transformation path (Proof) discovered by our method, and a brief analysis.

---

**EPLA-ProofNet #67**

**NL:** If $r$ is rational ($r \neq 0$) and $x$ is irrational, prove that $rx$ is irrational.

**Label:**
```
theorem exercise_1_1b (x : ℝ) (y : ℚ) (h : y ≠ 0) : (
Irrational x ) → Irrational ( x * y ) := by sorry
```

**Prediction:**
```
theorem mul_rat_tac_11959 (r : ℚ) (x : ℝ) (h : Irrational x) (
hr : r ≠ 0) : Irrational (r * x) := by sorry
```

**TED Similarity:** 0.23809523809523814
**TransTED Similarity:** 1

**Proof given by TransTED Similarity:**

---
```
example :
  (∀ (x : ℝ) (y : ℚ), y ≠ 0 → Irrational x → Irrational (x * ↑y
    )) =
    ∀ (r : ℚ) (x : ℝ), Irrational x → r ≠ 0 → Irrational (↑r *
      x) := by
  rw [forall_swap]
  apply forall_congr; intro _
  rw [← isRegular_iff_ne_zero']
  apply forall_congr; intro _
  rw [forall_swap]
  rw [@Rat.cast_comm] -- given by rw?
```
---

Comparing the label and prediction, human evaluator can easily match related variables and assumptions and give a correct judgment that they are semantically almost identical. However, TED Similarity soon gets into trouble dealing with lots of variable renaming and structure adjustment and gives out a really low similarity score. Fortunately, suggestions provided by `rw?` together with

additional tactics, such as `apply forall_congr; intro _` , make it possible to complete the natural matching step by step, leading to a more accurate estimation on semantic similarity.

---

**EPLA-miniF2F #170**

**NL:** A line $\ell$ passes through the points $B(7, -1)$ and $C(-1, 7)$. The equation of this line can be written in the form $y = mx + b$; compute $m + b$. Show that it is 5.

**Label:**
```
theorem mathd_algebra_142 (m b : ℝ) (h₀ : m * 7 + b = -1) (h₁ :
 m * -1 + b = 7) : m + b = 5 := by sorry
```

**Prediction:**
```
theorem my_favorite_theorem : let B : ℝ × ℝ := (7, -1); let C
: ℝ × ℝ := (-1, 7); ∀ m b : ℝ, (B.2 = m * B.1 + b ∧ C.2 = m *
C.1 + b) → m + b = 5  := by sorry
```

**TED Similarity:** $-0.03333333333333344$
**TransTED Similarity:** 1

**Proof given by TransTED Similarity:**

---

```
example :
(∀ (m b : ℝ), m * 7 + b = -1 → m * -1 + b = 7 → m + b = 5) =
(let B := (7, -1);
let C := (-1, 7);
∀ (m b : ℝ), B.2 = m * B.1 + b ∧ C.2 = m * C.1 + b → m + b = 5)
    := by
  apply forall_congr; intro _
  rw [mul_neg_one] -- given by rw?
  apply forall_congr; intro _
  rw [and_symm_left] -- given by rw?
  rw [and_symm_right] -- given by rw?
  rw [propext and_imp]
```

---

In this example, TransTED algorithm successfully matches corresponding variables and hypotheses by applying proper tactics and theorems, and gives a reliable evaluation result beyond structural likeness.

## F.4 FALSE POSITIVES IN DEFINITIONAL EQUALITY AND BEQ

This case study investigates the root cause of the false positives (FP) recorded by the Definitional Equality metric and the BEq metric, classified into three categories according to causes.

| Causes | miniF2F | ProofNet | Definitional Equality FP | BEq FP |
|---|---|---|---|---|
| Implicit Variable Matching | #24, #701 | #115, #128, #402 | + | − |
| Over-simplifying | #378, #603, #639, #641 | - | + | + |
| Header Mixing | - | #280 | + | + |

## F.4.1 IMPLICIT VARIABLE MATCHING

---

**EPLA-miniF2F #24**

**NL:** Expand the product $(x+1)^2 \cdot x$. Show that it is $x^3 + 2x^2 + x$.

**Label:**
```
theorem mathd_algebra_176 (x : ℝ) : (x + 1) ^ 2 * x = x ^ 3 + 2
 * x ^ 2 + x := by sorry
```

**Prediction:**
```
theorem my_favorite_theorem {R : Type*} [CommRing R] (x : R) :
 (x + 1) ^ 2 * x = x ^ 3 + 2 * x ^ 2 + x  := by sorry
```

**Definitional Equality**
```
example : mathd_algebra_176 = my_favorite_theorem := by rfl
```

---

In the Prediction, the statement was generalized by introducing an implicit type variable $\{\,R :$ `Type*`$\}$ with a typeclass assumption `[CommRing R]` instead of the concrete type $\mathbb{R}$, resulting in a semantic difference; however, Lean will automatically instantiate $R := \mathbb{R}$ and resolve the `CommRing` instance, so in Definitional Equality the compiler infers these instantiations and the two declarations become definitionally equal, thus passing the verifier and producing a false positive.

---

**EPLA-miniF2F #701**

**NL:** Expand $(x+3)(2x-6)$. Show that it is 2x^2-18.

**Label:**
```
theorem mathd_algebra_432 (x : ℝ) : (x + 3) * (2 * x − 6) = 2 *
 x ^ 2 − 18 := by sorry
```

**Prediction:**
```
theorem test_problem {R : Type*} [CommRing R] (x : R) : (x + 3)
 * (2 * x − 6) = 2 * x^2 − 18 := by sorry
```

**Definitional Equality**
```
example : mathd_algebra_432 = test_problem := by rfl
```

---

The error type here is identical to that in EPLA-miniF2F #24.

---

**EPLA-ProofNet #115**

**NL:** Prove that if $\Omega = \{1, 2, 3, \ldots\}$ then $S_\Omega$ is an infinite group.

**Label:**
```
theorem exercise_1_3_8 : Infinite (Equiv.Perm ℕ) := by sorry
```

**Prediction:**
```
theorem infinite_of_infinite_card_aux {Ω : Type u_1} [Infinite
Ω] : Infinite (Equiv.Perm Ω) := by sorry
```

**Definitional Equality**
```
example : exercise_1_3_8 = infinite_of_infinite_card_aux := by
rfl
```

---

In the Prediction, the model introduced an implicit type variable $\{\Omega\ :\ $ `Type u_1`$\}$ with a typeclass assumption `[Infinite Ω]` instead of the concrete type $\mathbb{N}$; however, Lean can instantiate $\Omega := \mathbb{N}$ and use the existing `Infinite` $\mathbb{N}$ instance, so in Definitional Equality the compiler infers these instantiations and the two declarations become definitionally equal, thus passing the verifier and producing a false positive.

---

**EPLA-ProofNet #128**

**NL:** Prove that $x^4 + 4x^3 + 6x^2 + 2x + 1$ is irreducible in $\mathbb{Z}[x]$.

**Label:**
```
theorem exercise_9_4_2c : Irreducible\n  (X^4 + 4*X^3 + 6*X^2 +
 2*X + 1 : Polynomial ℤ) := by sorry
```

**Prediction:**
```
theorem irreducible_wilsons_poly : Irreducible (wilsons_poly :
ℤ[X]) := by sorry
```

**Definitional Equality**
```
example : exercise_9_4_2c = irreducible_wilsons_poly := by rfl
```

---

In the Prediction, an undefined variable wilsons_poly appeared instead of the polynomial given in the problem, resulting in a semantic difference. However, Lean will automatically interpret wilsons_poly as an implicit variable. Therefore, in Definitional Equality, the Lean compiler will automatically infer wilsons_poly as the polynomial $X^4 + 4X^3 + 6X^2 + 2X + 1$, thus passing the verification and leading to a false positive.

---

**EPLA-ProofNet #402**

**NL:** Let $f\colon X \to X$ be continuous. Show that if $X = [0,1]$, there is a point $x$ such that $f(x) = x$. (The point $x$ is called a fixed point of $f$.)

**Label:**
```
theorem exercise_24_3a [TopologicalSpace I] [CompactSpace I] (f
 : I → I) (hf : Continuous f) : ∃ (x : I), f x = x := by sorry
```

**Prediction:**
```
theorem test_problem (f : Set.Icc (0 : ℝ) 1 → Set.Icc (0 : ℝ)
 1) (hf : Continuous f) : ∃ x, f x = x := by sorry
```

**Definitional Equality**
```
example : exercise_24_3a = test_problem := by rfl
```

---

In the Prediction, the model explicitly uses the concrete type, the interval $[0,1]$, for both the domain and codomain of the function $f$. However, in the Label, the theorem is stated more abstractly for any type $I$ equipped with needed instances. While $[0,1]$ can be endowed with such structure (making it a valid candidate for $I$), the type $[0,1] \to [0,1]$ is not definitionally equal to the function type $I \to I$ for an abstract $I$. Therefore, the definitions are not considered identical by the kernel, the `rfl` tactic fails, and this case is correctly identified as a negative example.

### F.4.2 OVER-SIMPLIFYING

---

**EPLA-miniF2F #378**

**NL:** What is the units digit of the sum of the squares of the first nine positive integers? Show that it is 5

**Label:**
```
theorem mathd_numbertheory_3 : (Σ x in Finset.range 10, (x + 1)
 ^ 2) % 10 = 5 := by sorry
```

**Prediction:**
```
theorem my_favorite_theorem : (Σ i in Finset.range 9, (i + 1)
^2) % 10 = 5 := by sorry
```

---

These two theorems produce exactly the same numerical result. However, the natural language description specifies computing the squares of the first nine positive integers, while the Label computes the squares of the first ten positive integers. Although this does not change the final value, the semantics are different. Nevertheless, BEq reports that the two theorems are equivalent, which does not match the actual situation.

---

**EPLA-miniF2F #603**

**NL:** What is the units digit of the sum of the squares of the first nine positive integers? Show that it is 5

**Label:**
```
theorem mathd_numbertheory_3 : (Σ x in Finset.range 10, (x + 1)
 ^ 2) % 10 = 5 := by sorry
```

**Prediction:**
```
theorem test_problem : (Finset.sum (Finset.Icc 1 9) (fun i => i
^2)) % 10 = 5 := by sorry
```

---

These two theorems produce exactly the same numerical result. However, the natural language description specifies computing the squares of the first nine positive integers, while the Label computes the squares of the first ten positive integers. Although this does not change the final value, the semantics are different. Nevertheless, BEq reports that the two theorems are equivalent, which does not match the actual situation.

---

**EPLA-miniF2F #639**

**NL:** How many integers between 15 and 85 are divisible by 20? Show that it is 4

**Label:**
```
theorem mathd_numbertheory_12 : Finset.card (Finset.filter (fun
 x => 20 | x) (Finset.Icc 15 85)) = 4 := by sorry
```

**Prediction:**
```
theorem test_problem : (Finset.filter (fun n => 20 | n) (
Finset.range 85 \ Finset.range 16)).card = 4 := by sorry
```

---

Although the Label and Prediction produce the same numerical result, the Label is not consistent with the natural language: it additionally includes 15 and 85 in the computation. While this does not affect the numerical result, it is not semantically equivalent to the natural language. On the other hand, the Prediction is strictly equivalent to the natural language. Therefore, the Label and Prediction are not equivalent.

---

**EPLA-miniF2F #641**

**NL:** What is the sum of the units digits of all the multiples of 3 between 0 and 50? Show that it is 78

**Label:**
```
theorem mathd_numbertheory_447 : (Σ k in Finset.filter (fun x
=> 3 | x) (Finset.Icc 1 49), k % 10) = 78 := by sorry
```

**Prediction:**
```
theorem test_problem : ((Finset.range 17).sum (fun i => (3 * i)
 % 10)) = 78 := by sorry
```

---

Although the computations for `Label` and `Prediction` are the same, `Prediction` additionally includes 0 and 50, whereas the natural language statement refers to the numbers between 0 and 50, excluding 0 and 50. Therefore, `Prediction` does not match the natural language. However, `BEq` will consider `Label` and `Prediction` as equivalent.

### F.4.3 HEADER MIXING

---

**EPLA-ProofNet #280**

**NL:** Show that $\sin(\pi/12)$ is an algebraic number.

**Label:**
```
open Real
open scoped BigOperators
theorem exercise_12_12 : IsAlgebraic ℚ (sin (Real.pi/12)) := by
  sorry
```

**Prediction:**
```
theorem my_favorite_theorem : IsAlgebraic ℚ (Real.sin (π / 12))
  := by sorry
```

**Definitional Equality**
```
example : thm_P = thm_Q := by rfl
```

---

In the Prediction, $\pi$ is not defined, so the two theorems are not equivalent. However, if the headers are merged, then $\pi$ becomes defined and the two theorems become equivalent. Since BEq first merges the headers and then performs mutual provability checking, BEq concludes that the two theorems are equivalent, which does not match the actual situation.

### F.5 PROMPT TEMPLATES

This section presents the prompts employed during the autoformalization phase, utilizing Gemini-2.5-Pro (Comanici et al., 2025) and Qwen3-Max (Yang et al., 2025). Subsequently, we detail the prompts for the majority voting phase, which leverages InternLM2-Math-Plus-7B (Ying et al., 2024b) for back-translation and DeepSeek-V3.2-Exp (DeepSeek-AI, 2025) for the NLI consistency check.

---

**Prompt Template for Autoformalization**

Please autoformalize the following problem in Lean 4 with a header. Use the following theorem names: my_favorite_theorem.
{informal_statement}

Your code should start with
```Lean4
import Mathlib
```

You should only output the theorem statement in Lean 4 format, ending with `by sorry`. You should NOT output the proof.

---

**Prompt Template for Back-Translation**

[UNUSED_TOKEN_146]user

Convert the formal statement into natural language:

```lean
{formal_statement}
```[UNUSED_TOKEN_145]

[UNUSED_TOKEN_146]assistant

---

---

**Prompt Template for NLI Check**

Please check following two math problems is same or different? Please consider each statement in two problems, they are different if any statement is different. Please point out any differences you found. Please reply \*\*same\*\* or \*\*different\*\* in the final sentence with bold format.

Problem 1: {THM_1}

Problem 2: {THM_2}

---

