# OpenReview forum: "ASSESS: A Semantic and Structural Evaluation Framework for Statement Similarity"
_ICLR.cc/2026/Conference — ICLR 2026 Poster_

### Official Review · Reviewer_FqxJ · 2025-10-24

**Soundness:** 2
**Presentation:** 3
**Contribution:** 2
**Rating:** 4
**Confidence:** 4

**Summary:**

This paper proposes a similarity score metric to assess the semantic and structural correspondence between pairs of formalized statements in autoformalization. Specifically, the proposed TransTED method leverages tree edit distance (TED) on operator trees extracted from Lean, and further augments TED through transformations based on tactics applied to equality relations between statements. To evaluate TransTED, the authors construct the EPLA dataset, which consists of autoformalized formal statements collected from MiniF2F and ProofNet. Experimental results show that TransTED achieves higher accuracy and Kappa coefficient compared to several baseline metrics on the newly constructed EPLA dataset, and demonstrates greater robustness than the BLEU score.

**Strengths:**

The paper is generally easy to read and well-organized. I appreciate the formalization and clear logical flow of the methodology section. The proposed method is well motivated and intuitively explained. Experimental results show that it outperforms existing baselines by a reasonable margin in accuracy and is better aligned with manual evaluations, demonstrating its effectiveness.

**Weaknesses:**

The proposed evaluation metric extends the traditional TED score through several augmentations; however, in practice, the implementation primarily applies a limited set of Lean tactics to establish the equality of two formal statements, and relies on a curated subset of these tactics. When the tactics fail to prove equivalence, the method simply falls back to the standard TED computation. Thus, the approach can be viewed as augmenting or relaxing proof-based equivalence checking with TED-based similarity, rather than achieving full proof-based validation. As a result, the evaluation metric does not guarantee 100% precision, unlike BEq or identity matching.

I also find some aspects of the experiments a bit unclear (see questions below). In my opinion, definitional equality should yield 100% precision—the reported misclassified examples mostly involve implicit type variables, which could reasonably be interpreted as equivalent or easily resolvable by instantiation.

Finally, I would suggest avoiding the claim that “to our knowledge, this represents the largest and most comprehensive dataset dedicated to this evaluation task.” Several recent works have autoformalized and manually verified a larger number of statements than those included in EPLA, such as:

[1] Autoformalize Mathematical Statements by Symbolic Equivalence and Semantic Consistency

[2] FormalMATH: Benchmarking Formal Mathematical Reasoning of Large Language Models

[3] Improving Autoformalization Using Type Checking

**Questions:**

How are semantic provability and structural similarity jointly used to evaluate the actual equivalence between two formal statements in the experiments? Could the authors provide a few concrete examples illustrating how these two aspects interact during evaluation?

In the example mathd_algebra_132, the labeled statements appear to perform a one-step reasoning that directly substitutes the point coordinates into the line equation. I am curious whether current LLMs also perform such autoformalization. Moreover, if an LLM conducts multiple reasoning steps during autoformalization, would it still be considered equivalent under the proposed metric?

What is the average computation time of TED and TransTED during evaluation?

---

> ### Author Response · Authors · 2025-11-20
>
> We thank you for your thoughtful review and for recognizing that our work is "well-organized" with a "clear logical flow". We value your constructive feedback and address your specific concerns below.
>
>
> *We acknowledge that the original framework diagram may not have fully clarified the calculation mechanism of TransTED Similarity. To address this, we have updated the diagram in the revised manuscript. Below, we answer Q1 firstly by combining a concrete example to demonstrate how semantic provability and structural similarity interact to compute the metric.*
>
>
> >**Q1: Interaction of Semantic Provability and Structural Similarity**
>
> **A1:** We appreciate this question. TransTED unifies these two aspects through a multi-step heuristic search using TED as the heuristic function.
>
> - **Semantic Provability (The Search Implementation):** The search is implemented by performing a tactical proof on the equality of the two original propositions. This constitutes the transformation process described in the paper, ensuring that every step strictly preserves semantic provability.
>
> - **Structural Similarity (The Heuristic Function):** We use the TED calculated between the operator trees of the left and right sides of the equation as the guiding heuristic. This encourages the search to select proof tactics that make the two sides as syntactically similar as possible, directly reflecting structural similarity.
>
> Theoretically, the global minimum difference achievable between the left and right sides represents the substantial semantic difference between the two original propositions. However, finding this global minimum is often computationally intractable. Therefore, in practice, we employ a mixed termination condition (e.g., time limit, tree size limit, or proof completion) and select the minimum TED encountered during the search path to calculate the final similarity score.
>
> Here is an illustrative example mathd\_algebra\_142 selected from EPLA benchmark.
>
> ```
> theorem mathd_algebra_142 (m b : ℝ) (h₀ : m * 7 + b = -1) (h₁ : m * -1 + b = 7) : m + b = 5 := by sorry
>
> theorem my_favorite_theorem :
> let B : ℝ × ℝ := (7, -1)
> let C : ℝ × ℝ := (-1, 7)
> ∀ m b : ℝ, (B.2 = m * B.1 + b ∧ C.2 = m * C.1 + b) → m + b = 5  := by sorry
> ```
>
> We first connect the two statements using an equal sign. By trying several tactic commands, we select the statement with the smallest ted value after applying the tactic transformation. We repeat this process until a stopping condition is met. The following code only shows the search trace towards the eventually selected result with minimum TED.
>
> ```
> -- TED = 30
> theorem node_0 :
> (∀ (m b : ℝ), m * 7 + b = -1 → m * -1 + b = 7 → m + b = 5) =
> (let B := (7, -1);
> let C := (-1, 7);
> ∀ (m b : ℝ), B.2 = m * B.1 + b ∧ C.2 = m * C.1 + b → m + b = 5)
> := by
> apply forall_congr; intro _
>
> -- TED = 26
> theorem node_1 (a : ℝ) :
> (∀ (b : ℝ), a * 7 + b = -1 → a * -1 + b = 7 → a + b = 5) =
>     ∀ (b : ℝ), (7, -1).2 = a * (7, -1).1 + b ∧ (-1, 7).2 = a * (-1, 7).1 + b → a + b = 5 := by
> rw [mul_neg_one] -- given by rw?
>
> -- TED = 22
> theorem node_2 (a : ℝ) :
> (∀ (b : ℝ), a * 7 + b = -1 → -a + b = 7 → a + b = 5) =
>     ∀ (b : ℝ), (7, -1).2 = a * (7, -1).1 + b ∧ (-1, 7).2 = -a + b → a + b = 5 := by
> apply forall_congr; intro _
>
> -- TED = 22
> theorem node_3 (a a_1 : ℝ) :
> (a * 7 + a_1 = -1 → -a + a_1 = 7 → a + a_1 = 5) =
>     ((7, -1).2 = a * (7, -1).1 + a_1 ∧ (-1, 7).2 = -a + a_1 → a + a_1 = 5) := by
> rw [and_symm_left] -- given by rw?
>
> -- TED = 18
> theorem node_4 (a a_1 : ℝ) :
> (a * 7 + a_1 = -1 → -a + a_1 = 7 → a + a_1 = 5) =
>     (a * (7, -1).1 + a_1 = (7, -1).2 ∧ (-1, 7).2 = -a + a_1 → a + a_1 = 5) := by
> rw [and_symm_right] -- given by rw?
>
> -- TED = 16
> theorem node_5 (a a_1 : ℝ) :
> (a * 7 + a_1 = -1 → -a + a_1 = 7 → a + a_1 = 5) =
>     (a * (7, -1).1 + a_1 = (7, -1).2 ∧ -a + a_1 = (-1, 7).2 → a + a_1 = 5) := by
> rw [propext and_imp] -- no goals to be saved

---

> ### Author Response · Authors · 2025-11-20
>
> >**W1: Precision vs. Recall and Method Design**
>
> **A1:** We acknowledge that TransTED Similarity does not guarantee 100% precision. However, we wish to clarify that this is not a flaw in the implementation, but a deliberate design choice inherent to continuous similarity metrics.
>
> **1. Continuous Metric vs. Binary Classification.** TransTED Similarity outputs a continuous similarity score rather than a binary decision. To generate binary decisions, a threshold must be applied.
>
> - **Trade-off:** In any threshold-based system, there is an inherent trade-off between False Positives and False Negatives.
>
> - **Design Goal:** Unlike rigorous decision metrics like BEq (which guarantee 100% precision but suffer from extremely low recall due to prover brittleness), TransTED Similarity is designed as a fractionalized metric. Its goal is to maximize alignment with human judgment by balancing precision and recall, rather than strictly filtering for absolute logical provability.
>
> **2. The Utility of False Positives.** TransTED Similarity's False Positives are rarely meaningless. Instead, they are typically samples with "high semantic similarity but minor technical discrepancies," such as:
>
> - Consistency issues in variable naming or type.
>
> - Differences in implicit type parameters.
>
> - Missing a single minor hypothesis.
>
> In practical scenarios like human-in-the-loop evaluation or dataset pre-screening, these similar but not correct candidates are highly valuable. A metric that flags these as highly similar helps human experts locate and correct minor errors, whereas a strict prover like BEq would simply reject them, providing no feedback.

---

> ### Author Response · Authors · 2025-11-20
>
> >**W2: Precision of Definitional Equality**
>
> **A2:** Through a complete case study, we figure out that there are multiple types of causes for Definitional Equality False Positive.
>
> **1. Implicit Variable Instantiation.** We respectfully clarify that if proposition $Q$ requires instantiation to obtain a proposition equivalent to proposition $P$, then proposition $Q$ can be implied by, but probably not equivalent to, proposition $P$. Therefore, classifying these cases as False Positives is methodologically correct.
>
> Consider the specific example mathd\_algebra\_176 from the failure analysis of Definitional Equality, where `thm_P` is defined on $\mathbb{R}$ and `thm_Q` is defined on any Commutative Ring $R$:
> - id: miniF2F\#24
> - label: `theorem mathd_algebra_176 (x : ℝ) : (x + 1) ^ 2 * x = x ^ 3 + 2 * x ^ 2 + x := by sorry`
> - prediction: `theorem my_favorite_theorem {R : Type*} [CommRing R] (x : R) :\n  (x + 1) ^ 2 * x = x ^ 3 + 2 * x ^ 2 + x  := by sorry`
> - Mathematical Discrepancy: `thm_Q` asserts the polynomial identity holds for *any* commutative ring. `thm_P` asserts it holds for $\mathbb{R}$. While the general law implies the specific case ($Q \implies P$), the specific case does not logically entail the general law ($P \not\implies Q$).
> - False Positive Mechanism: What Definitional Equality judges by `rfl` is that
>   ```
>   theorem thm_P (x : ℝ) : (x + 1) ^ 2 * x = x ^ 3 + 2 * x ^ 2 + x := by sorry
>   theorem thm_Q {R : Type*} [CommRing R] (x : R) : (x + 1) ^ 2 * x = x ^ 3 + 2 * x ^ 2 + x := by sorry
>   example : thm_P = thm_Q := rfl
>   ```
>   The Definitional Equality metric judges them as identical only because Lean automatically instantiates the implicit variable $R$ with $\mathbb{R}$ to satisfy the type checker.
>
> In conclusion, since the metric flags two propositions as equal when they are logically distinct (one implies the other, but they are not equivalent), we maintain that penalizing this as a False Positive is the correct decision for a rigorous evaluation benchmark.
>
> **2. Over-Simplifying.** For data on elementary mathematics, `rfl` might over-simplify left-hand and (or) right-hand side, leading to artificial semantic difference shrinking. All the False Positives of this type are miniF2F \#378, \#603, \#639, \#641. Here is one example among them:
> - id: miniF2F \#378
> - NL: What is the units digit of the sum of the squares of the first nine positive integers? Show that it is 5
> - Label: `theorem mathd_numbertheory_3 : (∑ x in Finset.range 10, (x + 1) ^ 2) % 10 = 5 := by sorry`
> - Prediction: `theorem my_favorite_theorem : (∑ i in Finset.range 9, (i + 1)^2) % 10 = 5 := by sorry`
> - Semantic Difference: Label states that $1^2 + \cdots + 10^2 \equiv 5 \mod 10$, whereas Prediction stats that $1^2 + \cdots + 9^2 \equiv 5 \mod 10$.
> - False Positive Mechanism: Both Label and Prediction are correct and simple arithmetic equations that can be proved by `rfl`.
>
> **3. Header Mixing.**
>
> - id: ProofNet \#280
> - NL: Show that $\sin (\pi / 12)$ is an algebraic number.
> - Label:
>   ```
>   open Real
>   open scoped BigOperators
>   theorem exercise_12_12 : IsAlgebraic ℚ (sin (Real.pi/12)) := by sorry
>   ```
> - Prediction:
>   ```
>   theorem my_favorite_theorem : IsAlgebraic ℚ (Real.sin (π / 12)) := by sorry
>   ```
> - Semantic Difference: In Prediction, $\pi$ is undefined and thus treated as an implicit variable of type $\mathbb{R}$. Therefore, Prediction actually states that for any real number $x$, $\sin (x/12)$ is algebraic over $\mathbb{Q}$. However, in Label $\pi$ is defined since because of these `open` commands.
> - False Positive Mechanism: Definitional Equality tries to prove by `rfl` the following equality:
>   ```
>   open Real
>   open scoped BigOperators
>   theorem thm_P : IsAlgebraic ℚ (sin (Real.pi/12)) := by sorry
>   theorem thm_Q : IsAlgebraic ℚ (Real.sin (π / 12)) := by sorry
>   example : thm_P = thm_Q := by rfl
>   ```
>   In this way, rfl proves the equality because the header is shared and $\pi$ in `thm_Q` is now defined in the usual way. In other words, the exact Lean expression is changed due to shared headers.

---

> ### Author Response · Authors · 2025-11-20
>
> >**W3: Dataset Claims**
>
> **A3:** We sincerely thank you for highlighting these important recent works. We agree that the claim regarding dataset size was imprecise given these concurrent developments. We have removed the claim that EPLA is the largest dataset and also added citations to [1], [2], and [3] in Related Work section.
>
>
> >**Q2: Comparison with One-step/Multi-step Reasoning**
>
> **A2:** As discussed, our algorithm involves multi-step reasoning, gradually attempting transformations to align the structural representation. We offer the following clarifications regarding the difference between one-step and multi-step reasoning in our framework:
>
> - **Human vs. Lean Granularity:** What appears to be a single step of substituting coordinates in human proof writing often requires a series of distinct commands in Lean to be rigorously verified. Our metric operates at this granular, tactical level rather than at the level of high-level human intuition.
>
> - **Technical Implementation:** In practice, our search is not implemented as a one-step jump. Instead, we extract the new goal state after each tactic command is applied. These intermediate states are then added to a priority queue for the next iteration of the search. Thus, the algorithm is inherently multi-step, building a transformation path tactic-by-tactic.
>
> - **Feasibility of LLM Integration:** Regarding your question about using an LLM to provide proof commands, this is entirely feasible within our framework. The system could support an LLM providing single or even multiple lines of commands at once. Theoretically, this would not disrupt the workflow, as the TED would continue to serve as the heuristic function guiding the search.
>
> - **Rationale for Exclusion:** However, we deliberately chose not to adopt an LLM-driven approach in this work, primarily for two key reasons. First, we aim to ensure the determinism and reproducibility of the TransTED Similarity metric. Relying on an LLM-based method for tactical generation would introduce non-deterministic behavior, which contradicts our emphasis on strict reproducibility. Second, we are concerned that an LLM might not preserve the original semantic difference between the two sides of an equation during proof generation. For instance, it could independently prove each side of a true equation, thereby converting the difference evaluation task into assessing the difference between both sides of `True = True`. Such a transformation process artificially shrinks the actual semantic difference and would interfere with a faithful evaluation.
>
>
> >**Q3: Computation Time Analysis**
>
> **A3:** We provide the detailed time statistics for all metrics in the table below. To ensure a fair comparison of resource requirements, we specify the hardware used:
>
> - **CPU Environment.** Dual Intel Xeon Silver 4216 CPUs (32 physical cores).
>
> - **GPU Environment.** One rented NVIDIA A800 (80GB) GPU.
>
> | Metric | EPLA-miniF2F (Avg Time seconds) | EPLA-ProofNet (Avg Time seconds) | Hardware Used |
> | :--- | :--- | :--- | :--- |
> | **(Baselines)** | | | |
> | Identity Match | ~0.00s | ~0.00s | Intel Xeon 4216 |
> | BLEU | ~0.01s | ~0.01s | Intel Xeon 4216 |
> | Majority Voting | 13.65s | 12.91s | NVIDIA A800 |
> | Definitional Eq. | 0.26s | 0.28s | Intel Xeon 4216 |
> | BEq | 31.45s | 35.75s | NVIDIA A800 |
> | **(Ours)** | | | |
> | TED Similarity | 1.20s | 1.32s | Intel Xeon 4216 |
> | TransTED Similarity** | 50.58s | 130.63s | Intel Xeon 4216 |
>
> - **TED Similarity (~1.2s)**: This metric is highly efficient on CPU. The time is primarily dominated by the overhead of the Lean Language Server (parsing and constructing the Operator Tree).
>
> - **TransTED Similarity (~50s - 130s).** The increased latency reflects the depth of the heuristic search on the CPU. TransTED explores a vast space of tactical transformations to find the minimal distance. The higher average time on EPLA-ProofNet (130s) compared to miniF2F (50s) reflects the greater structural complexity of undergraduate-level problems, which require deeper search paths to align.
>
> - **Efficiency vs. Accessibility Trade-off.** We acknowledge that TransTED has a higher wall-clock time per sample compared to GPU-accelerated baselines and will discuss it in the limitation section. However, we also want to emphasize that computational efficiency must be viewed through the lens of hardware accessibility. Out TED and TransTED eliminates the dependency on scarce GPU resources, making the evaluation framework significantly more accessible and cost-effective for the broader research community.
>
>
> We hope these clarifications and the updated diagram effectively address your concerns. We remain available for any further discussion.

---

> > ### Comment · Reviewer_FqxJ · 2025-11-21
> >
> > Thank you for the clarification. It improves my understanding of the paper and addresses many of my earlier concerns. I have adjusted my score accordingly. I also suggest that the authors include these examples and discussions in the revised manuscript, as they would make the presentation much clearer.
> >
> > I have one remaining question. The paper presents ablation studies on the thresholds used in TransTED, but it is not clear which threshold value(s) were applied for the results reported in Table 3. Could the authors specify the exact threshold(s) used in those experiments?

---

> > > ### Author Response · Authors · 2025-11-24
> > >
> > > >**Q4: Choice of Thresholds**
> > >
> > > **A4:** Thank you for increasing your score. We appreciate this additional question, as it significantly helps us improve the clarity of our experimental details.
> > >
> > > For all threshold-dependent metrics (BLEU, TED Similarity, and TransTED Similarity), the results reported in Table 3 represent the optimal performance (maximum Cohen's Kappa) achieved by sweeping the decision threshold. In the revised paper, we have updated Figure 3 to explicitly label these optimal points. The exact thresholds that yielded the reported Kappa scores are detailed below.
> > >
> > > | Metric | EPLA-miniF2F(Threshold, Max Kappa) | EPLA-ProofNet(Threshold, Max Kappa) |
> > > | :--- | :--- | :--- |
> > > | BLEU | (0.19, 0.26) | (0.35, 0.18) |
> > > | TED Similarity | (0.35, 0.31) | (0.30, 0.23) |
> > > | TransTED Similarity | (0.61, 0.35) | (0.38, 0.30) |
> > >
> > > We also have added a detailed discussion on threshold selection in Appendix F.
> > >
> > > ---
> > >
> > > We observe that the choice of threshold is inherently benchmark-dependent. This dependency stems from the richness of the underlying mathematical formalism:
> > >
> > > _plentiful mathematical structures $\Rightarrow$ more structurally variant expressions of the same meaning $\Rightarrow$ lower possible structural likeness_.
> > >
> > > For instance, the optimal TransTED Similarity threshold for EPLA-miniF2F (0.61) is significantly higher than that for EPLA-ProofNet (0.38). This disparity reflects the greater structural complexity and variation inherent to the latter. Consequently, we recommend that practitioners calibrate thresholds according to the structural diversity of their specific domain, using the optimal values reported in this work as reference baselines. To ensure robustness, we further advise evaluating model performance across a spectrum of thresholds rather than relying on a single static cut-off.
> > >
> > > ---
> > >
> > > >**Summary and Acknowledgements**
> > >
> > > Finally, we would like to thank you again for your review suggestions. Your thoughtful and professional feedback greatly helped us improve the paper. We have summarized updates in the paper corresponding to the weaknesses you proposed or the questions you raised.
> > >
> > > | Weakness/Question | Summary of Revision | Modified/Added Sections |
> > > | :--- | :--- | :--- |
> > > | **W1** (Precision of TransTED Similarity) | Explained trade-off between precision/recall. |**Appendix B**|
> > > | **W2** (Precision of Definitional Equality) | Analyzed three causes of false positives. | **Appendix F.4** |
> > > | **W3** (Dataset Claims) | Removed "largest dataset" claim; added citations to concurrent works. | **Section 1 & Section 2**|
> > > | **Q1&Q2** (Mechanism) | Updated the framework diagram | **Figures 1** |
> > > | **Q3** (Time Analysis) | Added detailed computation time and used hardware. | **Section 4.2 & Appendix F.2** |
> > > | **Q4** (Thresholds) | Added values of optimal thresholds and discussion on threshold selection. | **Figure 3 & Appendix F.1**|
> > >
> > > The updated paper has been uploaded to OpenReview and is available for viewing. If you have any further suggestions, we remain available for any further discussion during the discussion period.

---

### Official Review · Reviewer_nDGg · 2025-10-30

**Soundness:** 3
**Presentation:** 3
**Contribution:** 3
**Rating:** 2
**Confidence:** 5

**Summary:**

ASSESS is a two-stage framework to evaluate similarity between formal statements by combining structure and semantics. It parses statements into Operator Trees, then scores them with TransTED Similarity—an enhanced tree edit distance that applies semantic-preserving transformations. The authors introduce EPLA, a 524-pair expert-annotated benchmark (from miniF2F and ProofNet) labeled for both semantic provability and structural likeness. On EPLA, TransTED achieves state-of-the-art accuracy and the highest Cohen’s Kappa, outperforming string-, syntax-, proof-, and LLM-judge baselines. It is CPU-only, reproducible, provides continuous similarity scores, and ablations show the transformation component drives the gains.

**Strengths:**

- The proposed TransTED metric effectively combines syntactic and semantic information, overcoming the major limitation of existing purely structural or proof-based methods.
- Introduces a semantically enhanced Tree Edit Distance that accounts for logical transformations, providing graded similarity instead of binary judgments.
- Builds the largest expert-annotated dataset for this task (524 pairs), allowing rigorous and reproducible evaluation.

**Weaknesses:**

- No source code is provided.
- Why not compare with models like Kimina-Autoformalizer-7B and Goedel-Formalizer-V2-8B?
- Perhaps you could compare the recall accuracy with some embedding-based retrieval systems.

**Questions:**

Please refer to Weaknesses.

---

> ### Author Response · Authors · 2025-11-20
>
> We appreciate your positive assessment of the soundness, presentation, and contribution of our work, specifically your recognition that TransTED "effectively combines syntactic and semantic information" and that EPLA allows for "rigorous and reproducible evaluation". We address your specific concerns below.
>
>
> >**W1: No Provided Source Code**
>
> **A1:** We thank you for emphasizing the importance of code availability for reproducibility. As highlighted in our Global Response, we have now released the complete source code to fully address this concern.
>
> The anonymized repository is accessible here: **https://anonymous.4open.science/r/ASSESS-1103/**.
>
> This repository includes:
>
> 1. The full implementation of the TransTED Similarity metric.
>
> 2. The expanded EPLA benchmark.
>
> 3. Scripts to reproduce all reported experiments and results.
>
>
> >**W2: Comparison with Kimina/Goedel Autoformalizers**
>
> **A2:** We appreciate you raising this point, as it allows us to clarify the distinct roles of "Autoformalizers" versus "Evaluation Metrics" in our study, and to highlight how we have integrated the suggested models into our expanded benchmark.
>
> **1. Clarification of Scope: Metric vs. Generators.** It is important to distinguish between the metric used to measure performance and the generators being measured:
>
> - **Our Contribution (The Metric):** TransTED Similarity is an automated evaluation metric designed to judge the semantic correctness of the model-generated formal statements. Its competitors are other metrics like BLEU, LLM-as-a-judge and BEq.
>
> - **The Models Mentioned (The Generators):** Kimina-Autoformalizer and Goedel-Formalizer are autoformalization models. In our experimental setup, these models serve as the subjects to be evaluated, meaning our metrics assess whether their generated results are correct, rather than baselines to be compared against.
>
> Therefore, it is not methodologically valid to compare our metric TransTED Similarity against a generator.
>
> **2. Incorporating SOTA Models.** However, we fully agree with the spirit of the reviewer's comment: to ensure TransTED Similarity is robust, it must be tested on the outputs of state-of-the-art models. We have successfully addressed this by expanding our EPLA benchmark:
>
> -  **Inclusion of Goedel-Formalizer:** As detailed in our Global Response, we have generated formalizations using <u>Goedel-Formalizer-V2-8B</u> (and additionally <u>Qwen3-Max</u>) and incorporated them into the EPLA benchmark. This expanded the dataset from 524 to 1,247 pairs. Our re-evaluation on this expanded set confirms that TransTED Similarity remains the state-of-the-art metric, maintaining high agreement with human experts even on these advanced model outputs.
>
> - **Exclusion of Kimina Autoformalizer:** We deliberately excluded Kimina-Autoformalizer-7B from the benchmark expansion. This model utilized the miniF2F and ProofNet datasets during its training phase. Since EPLA is derived specifically from these test splits, evaluating a model on the exact data it was trained on would constitute data leakage, rendering the evaluation scientifically invalid.
>
>
> >**W3: Comparison with Embedding-based Retrieval Systems**
>
> **A3:** We appreciate the suggestion to compare against embedding-based retrieval systems. However, we would like to clarify a terminological distinction regarding "Recall" to explain why a direct comparison with retrieval systems is not methodologically applicable to our task.
>
> **1. Definition of "Recall" in Our Study.** In our experiment, we employ the standard definition of Recall for binary classification tasks:$$\text{Recall} = \frac{\text{True Positives (TP)}}{\text{True Positives (TP)} + \text{False Negatives (FN)}}$$
>
> This measures the metric's ability to correctly verify semantically equivalent pairs (a "True Positive" decision) out of all ground-truth equivalent pairs. Our task is **Semantic Evaluation**: determining if a specific generated statement is equivalent to the ground truth.
>
> **2. Distinction from Retrieval Tasks.** Embedding-based retrieval systems typically utilize "Recall@K" (ranking performance), which evaluates the system's ability to retrieve a correct document from a large database. Since our task involves judging a single specific pair rather than searching for a match among candidates (retrieval), the metrics are not directly comparable.
>
> **3. Comparable Baselines.** Instead of retrieval systems, we compared against baselines relevant to the evaluation task, including BLEU (string similarity), LLM-as-a-judge (semantic judgment), and BEq (mutual proof) as these share the same input-output structure and evaluation metrics (Accuracy/Kappa) as our proposed method.
>
>
> We hope our responses and the significant updates detailed in the Global Response, specifically the code release and benchmark expansion, have adequately addressed your concerns. We remain available to answer any further questions during the discussion period.

---

> > ### Comment · Reviewer_nDGg · 2025-11-26
> >
> > Thank you for the clarification. I’ve read your response carefully and now fully understand your points. The expanded dataset is meaningful.

---

> > ### Comment · Reviewer_nDGg · 2025-11-27
> >
> > In single-document retrieval, it’s completely feasible to split the document into chunks, encode them into representations, and then perform retrieval. Please consider adding some retrieval baselines based on representation models in the paper. Thank you for your efforts — I will raise my score.

---

### Official Review · Reviewer_qWzN · 2025-11-04

**Soundness:** 4
**Presentation:** 3
**Contribution:** 3
**Rating:** 6
**Confidence:** 3

**Summary:**

This work introduces ASSESS, a two stage evaluation framework that first translates natural language formal statements using the Lean Language Server into Operator and then measures the distance between the translation and a ground truth via a new metric TransTED. TransTED is a core contribution of the paper, and it extends from Tree Edit Distance by applying transformations to Operator Trees that surface semantic similarity (making the metric less prone to structurally different but semantically equivalent parses of formal statements). To prove the effectiveness of ASSESS the authors introduce a new dataset in this work, EPLA. EPLA is the other core contribution of this paper, and consists of 524 annotated pairs of natural language proofs and their formal translations annotated by experts.

The authors show that on EPLA, their method (ASSESS) consistently outperforms traditional evaluation metrics for autoformalization (this is a meta-evaluation, their evaluation framework is better at evaluating the task of auto-formalization).  The authors take care to compare against several common approaches for evaluating autoformalization including string based approaches, model-based approaches (like DeepSeek majority voting), and proof-based approaches.  ASSESS not only achieves higher accuracy, but has clear benefits to the alternative methods (including lower computational budget and reproducibility).  The authors compare their method by using a simple "Tree Edit Distance" (no translations for surfacing semantic similarity) and find that it leads to significant degradations when removed, showing the efficacy of their TransTED metric.

**Strengths:**

- **A large contribution: a dataset benchmarking evaluators for autoformalization + a method for evaluations**, a lot of work has gone into this paper, and it shows. The dataset EPLA is a nice contribution to the field for researchers working on evaluating and measuring the effectiveness of auto-formalization methods.
- **Strong main results: the leading approach behind theirs requires large amounts of compute (deepseek)**, excluding their ablated method, in order to achieve both high precision and recall when evaluating auto-formalization, their method appears to be the best as well as the most cost-effective.
- **Well written and clear results**, the math is a bit dense, but the paper overall is quite clear and well written. The graphs and tables show the results of their experiments clearly; figures 1 and 2 also help make their contributions a bit more concrete (showcasing an overview of the ASSESS framework and an example of an Operator Tree, respectively).

I lack the expertise to critically evaluate if the math theorems and proofs introduced in the paper are accurate or well-constructed, but the prose surrounding them was clear enough for me to understand and follow. Overall, this paper seems like a good contribution.

**Weaknesses:**

- **A lack of analysis** The method introduced does perform the best over all other evaluation methods; however, there is no discussion on the failure modes of the proposed method.  This makes it a bit difficult to understand where improvements could be made or more research is necessary (the EPLA dataset has some room left for improvement, so this may be a nice addition to the work).  *It is possible that some of this is discussed in the Appendix, and I missed it; if so, I'd still like a callout in the main body.*
- **Lack of ablations** There is one ablation, that of removing all the translations and using "TED" as the metric in ASSESS, but I do wonder how much of the gain TransTED gets from having each individual translation.  It may be worrisome if most of the gain from TransTED came from only a few of the translations in Table 1, or conversely, it may be insightful to see that all the translations improved the performance of TransTED, indicating that maybe future researchers could focus on devising more of these in order to push performance on EPLA further.  I think what I'd like to see is more discussion on whether the translations introduced provide the most gain and other translations won't help that much, or if adding a whole library of translations could improve performance significantly.


---

Minor notes:
- There's a weird artifact on lines 293 or 294 where the top of the text is slightly cut off.

**Questions:**

- Why is there no mutually provable & like* label? (I may be misunderstanding the relation between mutually provable and structurally similar under transformation)

---

> ### Author Response · Authors · 2025-11-20
>
> We appreciate your insightful suggestions, which have been instrumental in refining our paper. We address your specific questions below.
>
>
> >**W1: Failure Mode Analysis**
>
> **A1:** We sincerely thank you for this insightful suggestion. We agree that analyzing failure modes is essential for understanding the metric's boundaries and guiding future research. In response, we have added a dedicated Limitations and Future Work section. We categorize the primary failure modes into two types:
>
> **1. Limited Tactic Commands (False Negatives).** TransTED Similarity relies on a curated set of tactic commands (specifically rw? and 7 structural tactics). In complex cases requiring creative leaps or intermediate lemma synthesis, the heuristic search may fail to find a path, leading to a low score for equivalent statements.
>
> - Example EPLA-miniF2F #25:
>     - Label: `theorem mathd_numbertheory_353 (s : ℕ) (h₀ : s = ∑ k in Finset.Icc 2010 4018, k) : s % 2009 = 0 := by sorry`
>     - Prediction: `theorem my_favorite_theorem : (Finset.sum (Finset.Icc 2010 4018) id) % 2009 = 0  := by sorry`
>     - TED Similarity: 0.17
>     - TransTED Similarity: 0.17
>     - Failure Analysis: The current tactic set could not bridge the semantic gap between the variable definition s in the hypothesis and its direct usage in the conclusion.
>     - Future Work: This highlights the need to further optimize the configuration of tactic commands.
>
> **2. Semantic-Structural Tension (False Positives).** Because TransTED Similarity includes a structural similarity component, it can assign high scores to statements that are strxzucturally similar but semantically distinct (e.g., off-by-one errors or index shifts).
>
> - Example EPLA-ProofNet #360:
>     - Label: `theorem exercise_6_3 {n : ℕ} (a b : Fin n → ℝ) : (∑ i, a i * b i) ^ 2 ≤ (∑ i : Fin n, i * a i ^ 2) * (∑ i, b i ^ 2 / i) := by sorry`
>     - Prediction: `theorem test_problem (n : ℕ) (a b : Fin n → ℝ) : (∑ j : Fin n, a j * b j)^2 ≤ (∑ j : Fin n, (j.val + 1) * (a j)^2) * (∑ j : Fin n, (b j)^2 / (j.val + 1)) := by sorry`
>     - TED Similarity: 0.51
>     - TransTED Similarity: 0.81
>     - Failure Analysis: The final transformed statement is `theorem node_4 (a : ℕ) (a_1 a_2 : Fin a → ℝ) :(∑ i : Fin a, ↑↑i * a_1 i ^ 2) * ∑ i : Fin a, a_2 i ^ 2 / ↑↑i = (∑ j : Fin a, (↑↑j + 1) * a_1 j ^ 2) * ∑ j : Fin a, a_2 j ^ 2 / (↑↑j + 1) := by sorry`. The label (left hand side) iterates over $i$, while the prediction (right hand side) iterates over $j+1$. Mathematically, these are not equivalent in this context.
>     - Future Work: While this is a failure for binary verification, the searching result precisely locates the semantic divergence. This suggests that TransTED Similarity is highly effective for human-in-the-loop debugging, filtering nearly-correct candidates for expert review.

---

> ### Author Response · Authors · 2025-11-20
>
> >**W2: Ablation and Individual Tactic Contribution**
>
> **A2:** We appreciate this insightful suggestion. Understanding the individual contribution of each transformation is critical for guiding future research. We include a detailed analysis on the impact of these tactics into our paper, and here we provide a shorter version.
>
> **1. Challenge of Removal-Based Ablation.** We initially considered a removal-based ablation, i.e., removing one tactic at a time. However, because the tactics operate synergistically (e.g., `forall_congr` is often a prerequisite for `rw?` to find a rewrite inside a quantifier), removing one structural tactic can artificially zero-out the contribution of downstream semantic tactics.
>
> **2. Impact Analysis and Frequency.** Instead, to quantify the individual contribution of each tactic, we analyzed their usage on the expanded EPLA benchmark. The table below distinguishes between Search Process (how often the heuristic explorer attempted the tactic) and Final Transformation Path(how often the tactic appeared in the final transformation path that minimized the distance).
>
> | Tactic Command | Search Process| Final Transformation Path|
> | - |- |- |
> | apply congrArg | 38 | 19 |
> | apply congrFun | 18 | 9 |
> | apply forall_congr; intro _ | 908 | **347** |
> | apply implies_congr; all_goals (try rfl) | 67 | 6|
> | ext | 4 | 1|
> | rw [propext and_imp] | 64 | 22|
> | norm_cast | 1151 | 49|
> | rw? | **4579** | 285|
>
> - **The Power of `rw?`:** The generic library search `rw?` is the second most impactful tactic (285 uses). This confirms that leveraging the vast library of existing Lean lemmas is the most effective way to capture semantic equivalence.
>
> - **Structural Necessity:** `apply forall_congr` is the top contributor (347 uses). This indicates that before any semantic library search can happen, the metric *must* successfully navigate the quantifier structure of the statement.
>
> - **Future Direction:** Based on this, we suggest that future work should focus less on hand-curating specific rules (like `norm_cast`, which had lower yield relative to search cost) and more on optimizing **retrieval-based tactic generation** (like `rw?`) to tap into the full breadth of the mathematical library.
>
>
> >**Q1: The Label Settings**
>
> **A1:** Thank you for raising this detailed and insightful question. You are correct that the original label types were not sufficiently clear. To address this, we have modified the label format and provided a brief explanation in the Appendix for why only five distinct labels are used.
>
> **1. The New Label Format.** We have decomposed the label into three Boolean sub-labels. Each measures a specific type of similarity between pairs of formal statements and has its own meaning and application scenario:
>
> | Sub-label Name | Definition | Application Scenario |
> |:---------------|:-----------|:---------------------|
> | Provability | Whether the two statements are semantically equivalent. | To evaluate if a metric correctly assesses the semantic equivalence of two statements. |
> | Likeness before transformation | Whether the two (original) statements are structurally similar. | To evaluate if a metric reflects the editing cost required to make one statement structurally identical to another. |
> | Likeness after transformation | Whether semantically equivalent versions of the two statements can be made structurally similar. | To evaluate if a metric reflects the editing cost required to make statements semantically equivalent. |
>
> Among these, "likeness after transformation" is a novel criterion. Given current development trends, it has the potential to become an effective mode for interacting with code generators in formalizing mathematics. Therefore, alongside provability, this editing cost - which, from our perspective, is a form of likeness driven by practical formalization needs - serves as a crucial indicator of a formalization tool's capabilities.
>
> **2. Discussions on Value Combinations.** Due to inherent logical relationships between the sub-labels, only five value combinations are possible:
>
> | Provability | Likeness before transformation | Likeness after transformation |
> |:------------|:-------------------------------|:------------------------------|
> | true | true | true |
> | true | false | true |
> | false | true | true |
> | false | false | true |
> | false | false | false |
>
> The underlying logic is as follows:
> - **Provability implies likeness after transformation**, as no edits are needed to achieve semantic equivalence between the two propositions.
> - **Likeness before transformation implies likeness after transformation**, because the editing cost required to make statements semantically equivalent does not exceed that required to make them structurally identical.

---

### Official Review · Reviewer_18nk · 2025-11-09

**Soundness:** 3
**Presentation:** 3
**Contribution:** 3
**Rating:** 4
**Confidence:** 4

**Summary:**

This paper tackles a practical problem in automated theorem proving: how to automatically evaluate whether a machine-generated formal statement correctly captures the meaning of a natural language mathematical problem. The authors propose ASSESS, a framework that tries to balance both the structural form and semantic meaning when comparing formal statements.

**The approach works in two steps.**

First, they use the Lean Language Server to parse formal statements into operator trees (OPTs), tree structures that represent the syntactic hierarchy of the statement. They apply some preprocessing like adding placeholders and removing redundant parentheses to standardize these trees.

Second, they compute a similarity score using their proposed TransTED metric, which extends the classic Tree Edit Distance algorithm by incorporating semantic transformations. These transformations are implemented as Lean tactics (like `apply congrArg`, `ext`, etc.) that can recognize when two syntactically different statements are semantically equivalent. The algorithm uses heuristic search with TED as the guiding heuristic to find the minimum achievable distance.

**The paper makes three main contributions.**

- First is the TransTED Similarity metric itself, which outputs a continuous score between 0 and 1 rather than a binary decision. The authors provide theoretical justification through Theorem 1, which frames the problem in terms of pseudometric spaces.

- Second, they introduce EPLA, a new benchmark with 524 human-annotated statement pairs drawn from miniF2F and ProofNet. Each pair gets fine-grained labels covering both semantic correctness (provable vs unprovable) and structural similarity (like vs unlike), creating five categories in total.

- Third, they show experimental results where TransTED achieves 78.82% accuracy on miniF2F and 70.86% on ProofNet, with Kappa scores of 0.46 and 0.40, beating several baselines including BLEU, LLM-based voting, and proof-based methods.

The authors argue their method offers a sweet spot. It's more semantically aware than simple string matching, more robust than brittle proof-based approaches, and more efficient and reproducible than using LLMs as judges.

**Strengths:**

**Originality**

The paper makes several novel contributions. Applying operator trees to formal statement evaluation bridges mathematical information retrieval techniques to theorem proving in an unexplored way. The key innovation **augmenting Tree Edit Distance with semantic transformations via Lean tactics** elegantly addresses the challenge that semantically equivalent statements can have very different syntactic forms. The pseudometric space formulation (Theorem 1) provides theoretical grounding. The EPLA benchmark's five-category annotation scheme is notably more nuanced than existing binary evaluation datasets.

**Quality**

The technical approach is sound. Leveraging the Lean Language Server for OPT construction is pragmatic engineering. The choice of transformation tactics (Table 1) reflects good mathematical intuition. The experimental evaluation covers diverse baselines (string-based, proof-based, LLM-based). The ablation study provides clear evidence that transformations contribute meaningfully (+6.70% on miniF2F). Implementation details demonstrate attention to reproducibility.

**Clarity**

The paper is well-structured with logical progression from motivation through methodology to experiments. Mathematical notation is introduced systematically with helpful examples. Figure 3 effectively demonstrates TransTED's robustness to threshold selection compared to BLEU's sensitivity. Results are presented with appropriate metrics and clear tables.

**Significance**

The work addresses a real bottleneck where existing evaluation methods have well-documented limitations. An efficient (CPU-only), reproducible automated metric could significantly accelerate research by enabling faster iteration without expensive human evaluation. Outperforming LLM-based majority voting while being much cheaper suggests genuine practical utility. The EPLA benchmark could become a standard evaluation resource for the community.

**Weaknesses:**

**1. Missing Citation of Highly Related Concurrent Work [Major - Critical for Novelty Assessment]**

The paper does not cite GTED (Liu et al., "Generalized Tree Edit Distance: A Faithful Evaluation Metric for Statement Autoformalization", arXiv preprint, arXiv:2507.07399v1, July 10, 2025), despite substantial overlap in core contributions. Both works use operator trees extracted via Lean Language Server, extend Tree Edit Distance with semantic transformations, and evaluate on miniF2F and ProofNet. The timeline raises concerns: GTED appeared on arXiv on July 10, 2025, while the ICLR 2026 deadline was September 24, 2025, where a 2.5-month gap that should have been sufficient for discovery and citation, especially given the direct relevance to this work.

Without proper comparison, several critical questions remain unanswered. I cannot assess whether the pseudometric formulation (Theorem 1) is substantially different from GTED's "generalized tree transformation" framework, or whether they are merely different mathematical presentations of the same underlying idea. The transformation sets appear different quantitatively (ASSESS: 6-7 tactics vs GTED: primarily α-conversion), but it's unclear if this represents a genuine methodological advance or simply different implementation choices. Most importantly, the reported performance differences (GTED: 70.73% accuracy on miniF2F vs ASSESS: 78.82%) cannot be properly interpreted without knowing whether they stem from algorithmic improvements or merely different evaluation datasets.

If the authors were genuinely unaware of GTED during development, this represents an unfortunate collision of independent work and should be acknowledged. However, if they were aware but chose not to cite it, this falls into more serious issues of academic integrity. Either way, the current manuscript must address this relationship. I believe the contribution lacks sufficient demonstrated novelty until the authors: (1) explicitly cite GTED and clarify their awareness timeline, (2) provide detailed technical comparison showing what is genuinely new beyond GTED, and (3) ideally, evaluate both methods on identical test sets to enable fair performance comparison.

I believe this issue significantly impacts my assessment of the paper's originality and must be resolved for acceptance.

---
**2. Dataset weakness**

2.1 Selection bias

The benchmark includes samples that successfully compile after translation, creating severe selection bias. Based on GTED's reported numbers using Herald translator alone, the compilation success rates are approximately 84% for miniF2F (205/244) and only 25% for ProofNet (93/371). The ProofNet success rate is particularly concerning, the evaluation excludes 75% of the original test set, precisely the cases where translation is most challenging.

While ASSESS uses both Herald and Gemini translators (producing 373 miniF2F pairs and 151 ProofNet pairs as shown in Table 2), the underlying selection bias remains: the metric is only evaluated on cases where at least one translator produced compilable output. We have no information about how TransTED would behave on the harder cases where translation completely fails. This severely limits conclusions about the metric's general applicability.

2.2 Translator bias

The benchmark relies exclusively on two translators (Herald and Gemini 2.5 Pro). Different autoformalization systems may produce systematically different output patterns, where some favoring verbose formulations, others producing compact statements, or using different Mathlib conventions. TransTED's performance might not transfer to outputs from other translation systems. Including diverse translators would strengthen claims about the metric's general applicability. This selection bias severely limits the conclusions we can draw about the metric's general applicability. The paper should explicitly discuss this limitation and its implications.

2.3 Simple statement-only evaluation

Both miniF2F and ProofNet contain only simple, single-statement theorems , and typically one-line formulas expressing direct mathematical relationships. Real world formalization is substantially more complex, often involving custom definitions (def), structured types (structure, inductive), multi-statement developments with auxiliary lemmas, and nontrivial type hierarchies.

For example, a predicted formalization might define a helper function that accomplishes the same mathematical purpose as the ground truth's , but with different naming and internal structure. The current OPT-based approach, which only compares the final theorem statement in isolation, might struggle with such cases. The paper provides no discussion of this limitation. It remains unclear whether TransTED can handle complex multi-component formalizations, whether OPT construction can parse and align definitions appropriately, or whether transformation tactics generalize beyond simple theorems.

**Questions:**

**Q1: Relationship with GTED**

As noted in the weaknesses, GTED has substantial overlap with your work. This is the most critical issue affecting our assessment:

- Can you clarify whether you were aware of GTED during development?
- Please provide detailed technical comparison showing what is genuinely new in your approach
- Would you be willing to evaluate both methods on identical test sets for fair comparison?

Your response will directly determine whether we view this as independent innovation or incremental refinement of GTED.

---

**Q2: Handling complex formalizations [Important - could mitigate scope limitation]**

The evaluation focuses on simple, single-statement theorems. We understand this reflects the datasets used, but it raises questions about practical applicability.

- Have you conducted any experiments (even informal) with formalizations containing definitions, structures, or multi-statement developments?

- Can you discuss theoretically how the OPT construction and transformation tactics would handle such cases?

Even preliminary evidence or principled discussion would help assess whether the scope limitation is fundamental or merely reflects current evaluation constraints.

---

**Q3: Transformation tactics design [Moderate - would strengthen contribution]**

Table 1 presents seven transformation tactics. Can you provide more details like

- How were these tactics selected? Systematic exploration or intuition?
- Have you analyzed their individual contributions (ablation study)?
- Which tactics are most frequently used or impactful?
This would help understand the design rationale and completeness of the transformation set.

---

**Q4: Statistical reliability of improvements**

For ProofNet (151 samples), the +3.31% improvement represents only about 5 additional correct predictions. Have you checked whether this difference is statistically significant? A simple test would help confirm the reliability of this improvement, though the miniF2F results (+6.70% on 373 samples) are more clearly substantial.

---

> ### Author Response · Authors · 2025-11-20
>
> We sincerely appreciate your comprehensive summary and positive assessment of our work. We address your specific questions and concerns below.
>
>
> >**W1 & Q1: Relation to GTED and Novelty**
>
> **A1:** We thank you for identifying the connection to GTED. To adhere to ICLR's double-blind policy, we address the relationship without disclosing authorship, clarifying the status and differences to resolve the novelty concern:
>
> **1. Status of GTED vs. ASSESS.** We wish to clarify that **GTED is a preliminary, non-archival workshop paper** presented at the ICML 2025 AI4Math Workshop. As per the workshop's policy, it does not constitute prior archival publication. The current paper, ASSESS, represents the rigorous formalization and fully realized evolution of that preliminary exploration. We will explicitly cite GTED and clarify this relationship in the final revision.
>
> **2. Technical Advancements.** The performance gap stems from fundamental algorithmic advancements, not merely implementation differences. ASSESS introduces three critical optimizations that differentiate it from the preliminary GTED implementation:
>
> - **Formal Theoretical Framework:** While GTED operated on heuristic "tree transformations," ASSESS formalizes these concepts into a rigorous pseudometric space with provable properties, providing the theoretical guarantee that was absent in the preliminary work.
>
> - **Advanced Semantic Mechanism:** GTED relied primarily on simple **$\alpha$-conversion** (variable renaming) to handle semantic nuances. In sharp contrast, ASSESS introduces a **proof-based transformation approach**. Our method intelligently concatenates propositions to perform context-aware transformations, enabling deep semantic alignment (e.g., recognizing logical equivalences beyond renaming) before calculating the Tree Edit Distance. This is a qualitative leap in methodology, not just a parameter tune.
>
> - **Code Bug Fixes:** The preliminary code in GTED suffered from rule-matching bugs within the Lean Language Server. ASSESS resolves these issues, resulting in significantly higher stability and reproducibility.
>
> In conclusion, ASSESS is to GTED what a finalized conference paper is to an early workshop abstract: it provides the rigorous theory, the advanced algorithm (proof-based vs. renaming), and the stable implementation necessary for a standard benchmark.
>
>
> >**W2.1: Selection Bias**
>
> **A2.1:** We respectfully clarify that the exclusion of uncompilable samples is not a statistical selection bias, but a necessary methodological prerequisite for **semantic evaluation**.
>
> In the standard autoformalization evaluation pipeline, verification follows a strict hierarchy:
>
> **1. Syntactic Check (Compiler).** The system first checks if the generated code compiles. If it fails, the generation is demonstrably incorrect. No advanced metric is needed to determine this.
>
> **2. Semantic Check (Our Focus).** If the code compiles, the core research problem is to determine if it mathematically matches the informal statement. This is where TransTED Similarity contributes value.
>
> On the other hand, including uncompilable noise would artificially inflate the performance of all metrics (as they would all easily reject broken code), effectively masking their ability to distinguish subtle semantic differences in valid code.
>
> Therefore, to strictly evaluate the discriminative ability of the metrics, our benchmark retains only the samples that have passed compilation. This ensures we are measuring the metric's ability to detect semantic misalignment, rather than simply counting syntax errors.
>
>
> >**W2.2: Translator Bias**
>
> **A2.2:** We agree that evaluating diverse output styles is critical for generalization. To address this, we have significantly expanded our benchmark to minimize translator bias and directly test the metric against varied formalization patterns.
>
> **1. Benchmark Expansion.** As detailed in our Global Response, we have scaled the evaluation from two to four distinct models, specifically selecting them to cover the spectrum of "domain-specific vs. general-purpose" selections:
>
> - **Domain-Specific Models:** Herald Translator and Goedel-Formalizer-V2-8B. (Tend to follow stricter Mathlib conventions).
>
> - **General-Purpose LLMs:** Gemini 2.5 Pro and Qwen3-Max. (Tend to exhibit more linguistic variety and diverse verbosity levels).
>
> **2. Results and Robustness.** This expansion increases the dataset to **1,247 samples** (see Global Response for detailed breakdown). Crucially, our re-evaluation demonstrates that **TransTED Similarity maintains its state-of-the-art performance** across this augmented dataset. This confirms that our metric is not overfitted to a specific translator's style but is robust to the diverse output patterns produced by different model families.

---

> ### Author Response · Authors · 2025-11-20
>
> >**W2.3 & Q2: Statement-only Evaluation**
>
> **A2.3:** We acknowledge that real-world formalization often involves complex structures beyond single-line theorems. We address this from both a benchmark perspective and a technical perspective.
>
> **1. Alignment with Community Standards.** Our work targets the specific task of "Statement Autoformalization", i.e. translating natural language statements into formal statements. In this specific domain, miniF2F and ProofNet are the established, standard benchmarks used by recent works. Adhering to these benchmarks ensures that our metrics are more applicable and accurately reflect the true performance of current autoformalization models.
>
> **2. Context Awareness.** Critically, TransTED Similarity is not limited to isolated single-line statements. It is designed to handle the custom definitions and helper functions described by the reviewer through context loading:
>
> - **Handling Auxiliary Definitions:** When a generated formalization includes supplementary structures (e.g., def, structure, or class) prior to the theorem statement, TransTED Similarity does not discard them. Instead, it loads these definitions into the local Lean environment before constructing the equality for comparison.
>
> - **Semantic Equivalence:** Because these auxiliary definitions are present in the environment, the transformation tactics (e.g., rw) can unfold and utilize them during the proof search. Therefore, TransTED can successfully identify this equivalence via the semantic transformation stage, provided the auxiliary definitions are syntactically valid.
>
>
> >**Q3: Transformation Tactics Design**
>
> **A3:** We thank you for this suggestion and we include the detailed analysis on these tactics into the paper. Here we provide a short explanation of the rationale behind our tactic selection and a simple analysis of their impact.
>
> **1. Selection Rationale.** Our selection process combined theoretical coverage with empirical necessity to ensure robust semantic alignment:
>
> - **Systematic Selection:** The first five tactics were selected systematically to cover the fundamental properties of equality and logic. These allow the metric to traverse the structural hierarchy of the Operator Tree (e.g., moving inside quantifiers or function arguments).
>
> - **Empirical Selection:** The remaining tactics address specific, recurring friction points observed in Lean formalization:
>     - `norm_cast`: Specifically handles type coercion mismatches (e.g., $\mathbb{N}$ vs. $\mathbb{Z}$), which are common sources of "silent" semantic equivalence.
>     - `rw [propext and_imp]`: Addresses the structural difference between curried implication ($A \to B \to C$) and logical AND ($A \wedge B \to C$), ensuring these equivalent forms can be aligned.
>
> **2. Challenge of Removal-Based Ablation.** We initially considered a removal-based ablation, i.e., removing one tactic at a time. However, because the tactics operate synergistically (e.g., `forall_congr` is often a prerequisite for `rw?` to find a rewrite inside a quantifier), removing one structural tactic can artificially zero-out the contribution of downstream semantic tactics.
>
> **3. Impact Analysis and Frequency.** To quantify the individual contribution of each tactic, we analyzed their usage on the expanded EPLA benchmark. The table below distinguishes between Search Process (how often the heuristic explorer attempted the tactic) and Final Transformation Path(how often the tactic appeared in the final transformation path that minimized the distance).
>
> | Tactic Command | Search Process| Final Transformation Path|
> | - |- |- |
> | apply congrArg | 38 | 19 |
> | apply congrFun | 18 | 9 |
> | apply forall_congr; intro _ | 908 | **347** |
> | apply implies_congr; all_goals (try rfl) | 67 | 6|
> | ext | 4 | 1|
> | rw [propext and_imp] | 64 | 22|
> | norm_cast | 1151 | 49|
> | rw? | **4579** | 285|
>
> - **Most Impactful:** `apply forall_congr` is the dominant structural tactic (347 successful applications), highlighting that handling quantifiers is the most critical step in aligning formal statements.
>
> - **Most Versatile:** `rw?` is heavily utilized, proving that general mathematical rewriting using the Mathlib library is essential for capturing semantic equivalence beyond structural changes.
>
> - **High Exploration, Specific Yield:** `norm_cast` has a high search count but a lower final yield. This indicates that while type mismatches are frequent candidates for exploration, they only constitute the solution in specific, crucial edge cases.

---

> ### Author Response · Authors · 2025-11-20
>
> >**Q4: Statistical Reliability of Improvements**
>
> **A4:** We appreciate the reviewer's rigorous attention to statistical reliability. We agree that on the original small EPLA-ProofNet subset (N=151), the raw count of improved samples was limited. This limitation was a primary motivation for the significant benchmark expansion detailed in our Global Response.
>
> **1. Verification via Expanded Benchmark.** To confirm that these improvements are structural rather than statistical noise, we re-evaluated the metrics on the expanded EPLA-ProofNet, which increased the sample size from 151 to 416.
>
> On this dataset (nearly $3\times$ larger), TransTED Similarity maintains its superiority over the baseline TED Similarity, improving Accuracy from 64.67% to 67.31% and increasing the Kappa coefficient from 0.23 to 0.30.
>
> The fact that the performance advantage is sustained, and even more pronounced in terms of inter-rater agreement (Kappa), across a much larger and more diverse dataset confirms that the improvement is reliable and robust.
>
> **2. Deterministic Reproducibility.** Regarding the nature of the algorithm: We wish to clarify that TransTED is fully deterministic. Unlike methods relying on random sampling (e.g., LLM-as-a-Judge with temperature > 0), our algorithm relies on deterministic tactical applications and sorted heuristic search. Therefore, any improvements are fully reproducible values that result from the method's superior semantic alignment, not run-to-run variance.
>
>
> We thank you again for your constructive criticism, which has directly led to a stronger and more transparent submission. We hope these updates effectively resolve your concerns regarding novelty and robustness.

---

### Author Response · Authors · 2025-11-20
**General Response**

We thank the reviewers for their thoughtful and constructive feedback. Before addressing specific questions, we wish to highlight four major updates to our work that directly strengthen the reproducibility, scope, robustness, and clarity of the proposed evaluation framework.

**1. Release of Source Code**

To ensure full transparency and facilitate immediate reproducibility, we have open-sourced our complete project code. While our initial submission provided the benchmark and results in the supplementary materials, we have now organized the codebase into an anonymous repository for easier inspection and review: https://anonymous.4open.science/r/ASSESS-1103/

**2. Expansion of the EPLA Benchmark**

We have substantially scaled up the EPLA benchmark (increasing the dataset size from 524 to 1,247 pairs) by incorporating candidate outputs from two additional state-of-the-art models: **Goedel-Formalizer-V2-8B** and **Qwen3-Max**. Including these models introduces a broader diversity of formalization styles and error patterns. This ensures that EPLA is not biased toward specific model architectures and better represents the current landscape of autoformalization capabilities. The updated data distribution is shown below:

| Dataset         | Herald Translator | Goedel-Formalizer | Gemini 2.5 Pro | Qwen3-Max | Total |
|-----------------|-------------------|--------------------|----------------|-----------|-------|
| **EPLA-miniF2F**    | 198          | 234                | 175            | 224       | **831**   |
| **EPLA-ProofNet**   | 89           | 130                | 62             | 135       | **416**   |

**3. Evaluation on Expanded Data**

We re-evaluated TransTED Similarity and all competing baselines on this expanded benchmark. **Crucially, our results confirm the stability and generalizability of TransTED Similarity**. Despite the increased diversity and difficulty of the expanded dataset, TransTED Similarity maintains its state-of-the-art performance, securing the highest Accuracy and Kappa coefficients. This demonstrates that our metric is robust to varying generator distributions and is not overfitted to the original test set. Detailed experimental results are provided below:

| Metric | EPLA-miniF2F Precision | Recall | Accuracy | Kappa | EPLA-ProofNet Precision | Recall | Accuracy | Kappa |
|--------|--------------------------|-----------------------|--------------------------|----------------------|---------------------------|-------------------------|----------------------------|------------------------|
| (Baselines) | | | | | | | | |
| Identity Match | **100.00%** | 8.79% | 32.61% | 0.05 | **100.00%** | 3.69% | 43.51% | 0.03 |
| BLEU | 82.25% | **73.94%** | 68.96% | 0.26 | 72.30% | 43.85% | 57.21% | 0.18 |
| Majority Voting | 91.00% | 31.27% | 46.93% | 0.14 | 66.67% | 45.08% | 54.57% | 0.12 |
| Definitional Equality | 97.09% | 32.57% | 49.46% | 0.19 | 86.21% | 10.25% | 46.39% | 0.07 |
| BEq | 98.60% | 45.77% | 59.45% | 0.29 | 98.77% | 32.79% | 60.34% | 0.28 |
| (Ours) | | | | | | | | |
| TED Similarity | 84.51% | 71.99% | 69.56% | 0.31 | 66.11% | 81.56% | 64.67% | 0.23 |
| **TransTED Similarity** | 87.97% | 69.06% | **70.16%** | **0.35** | 68.49% | **81.97%** | **67.31%** | **0.30** |


**4. Updated Framework Diagram**

We have updated the framework diagram in the revised manuscript. The new diagrams explicitly illustrate the multi-step heuristic search process, clarifying how tactic transformations are guided by Tree Edit Distance to bridge the gap between semantically equivalent but structurally distinct statements.

---

### Meta-Review · Area_Chair_4VaS · 2026-01-07

**Summary:**

- **18nk** (4): Flagged missing citation to concurrent GTED work as critical novelty issue; questioned dataset selection/translator bias and limited evaluation to simple statements.
- **nDGg** (2 -> 6): Initially cited lack of source code and misunderstood scope (comparing metrics vs autoformalization models).
- **qWzN** (6): Requested failure mode analysis and ablation studies on transformation tactics.
- **FqxJ** (4 -> 6): Questioned implementation limited to curated tactics and definitional equality false positives.

**Reviewer Concerns:**

The authors actively replied to all raised issues/concerns.

Outstanding concern: Focus on single-statement theorems remains a limitation, though authors justify this aligns with community benchmarks. This would be a future work for the authors.

**Reviewer Scores:**

nDGg: 2 -> 6

FqxJ: 4 -> 6

---

### Decision · Program_Chairs · 2026-01-26

Accept (Poster)